# The p55TNFR-IKK2-Ripk3 axis orchestrates arthritis by regulating death and inflammatory pathways in synovial fibroblasts

Marietta Armaka [1], Caroline Ospelt[2], Manolis Pasparakis [3] & George Kollias[1,4]

NFκB activation and regulated cell death are important in tissue homeostasis, inflammation and pathogenesis. Here we show the role of the p55TNFR–IKK2l–Ripk3 axis in the regulation of synovial fibroblast homeostasis and pathogenesis in TNF-mediated mouse models of arthritis. Mesenchymal-specific p55TNFR triggering is indispensable for arthritis in acute and chronic TNF-dependent models. IKK2 in joint mesenchymal cells is necessary for the development of cartilage destruction and bone erosion; however, in its absence synovitis still develops. IKK2 deletion affects arthritic and antiapoptotic gene expression leading to hypersensitization of synovial fibroblasts to TNF/Ripk1-mediated death via district mechanisms, depending on acute or chronic TNF signals. Moreover, Ripk3 is dispensable for TNF-mediated arthritis, yet it is required for synovitis in mice with mesenchymal-specific IKK2 deletion. These results demonstrate that p55TNFR–IKK2–Ripk3 signalling orchestrates arthritogenic and death responses in synovial fibroblasts, suggesting that therapeutic manipulation of this pathway in arthritis may require combinatorial blockade of both IKK2 and Ripk3 signals.

[1] Biomedical Sciences Research Center "Alexander Fleming", 16672 Vari, Greece. [2] Center of Experimental Rheumatology, University Hospital Zurich and University of Zurich, Wagistrasse 14, Schlieren, 8952 Zurich, Switzerland. [3] Institute for Genetics, University of Cologne, 50674 Cologne, Germany. [4] Department of Physiology, Medical School, National and Kapodistrian University of Athens, 11527 Athens, Greece. Correspondence and requests for materials should be addressed to M.A. (email: armaka@fleming.gr) or to G.K. (email: kollias@fleming.gr)

Rheumatoid arthritis (RA) is an inflammatory disease that primarily affects the synovial joints, especially the fingers, wrists, feet, and ankles, resulting in painful deformities and immobility. RA has now been associated with multi-disease syndromes and comorbidities that affect bone and lipid metabolism, the cardiovascular, gastrointestinal and respiratory systems, and increased cancer susceptibility[1]. The socioeconomic burden of rheumatic diseases is thus amplified more than once thought. Regardless of the therapeutic advances provided by effective new drugs, absolute remission of disease has not been achieved. Further investigation of the aetiopathogenesis of RA is required.

Current understanding of RA pathogenesis is focused mainly on inflammation in the joints, which is maintained by the deregulated production of inflammatory mediators by immune and mesenchymal cell types that are fueled mainly by tumour necrosis factor (TNF)[2,3]. Functional genetic analyses in TNF-overexpressing mice ($hTNFtg$[4] and $TNF^{\Delta ARE}$[45]) highlight the fundamental role of synovial fibroblasts (SF) and p55TNFR in mediating arthritogenesis, demonstrating that TNF/p55TNFR signalling in SFs is sufficient to orchestrate proliferation, inflammation, and tissue destruction in these models[6]. In principle, p55TNFR signalling is driven by complex I and II, which consist of Tradd/Ripk1/Traf2/cIAP1 (I) and FADD/caspase-8/Ripk1 (IIa) or Ripk1/Ripk3/Mlkl (IIb). These complexes lead to survival and inflammatory signalling (such as nuclear factor κB (NFκB)), as well as death pathways that are reliant on the availability of NFκB and caspases (reviewed in ref. [7]).

Aside from apoptosis, necroptosis is another form of programmed cell death controlled by the p55TNFR pathway upon the suppression of both NFκB and caspase signalling and is dependent on the kinase activity of Ripk1 and the interaction of Ripk1/Ripk3/Mlkl (reviewed in ref. [8]). However, there are instances of TNF-induced necroptotic death happening in the presence of caspase machinery such as the case of L929 fibrosarcoma cell line[9]. Some other studies challenge the idea that NFκB/caspase inhibition is required for activation of Ripk1/Ripk3/Mlkl signalling and suggest that necroptosis induction is dependent on both the stimuli and the cell context. For example, Toll-like receptors (TLR) are equally effective in driving Ripk1/3-dependent and Mlkl-dependent necroptosis with different molecular prerequisites[10–13]; the kinase activity of Ripk1 is required for both p55TNFR-induced or TLR-induced necroptosis in macrophages[12,14,15], but this activity is dispensable for TLR-induced necroptosis in endothelial cells or fibroblasts[12].

Given the essential role of the NFκB pathway in the transcription of regulators of inflammatory and death responses and the identification of several risk loci for NFκB regulators in patients with RA[16–18], many approaches to NFκB inhibition have been used to understand the proinflammatory function of NFκB in vivo and ex vivo. Several studies have demonstrated the effect of NFκB inhibition in the activation of RA-SFs ex vivo[19–23], and preclinical therapeutic schemes including small molecule targeting of IKK2 or virus-based vectors expressing inhibitors of NFκB activation have been tested in RA models; therapeutic effects have been shown to mainly affect immune cell activation[24–28]. However, the potential induction of cell death resulting from NFκB inhibition and its contribution to arthritis is less studied[28]. Further understanding of the function of NFκB and SF death pathways in chronic inflammatory arthritis is necessary.

Here we dissect the requirement for mesenchymal p55TNFR and IKK2/Ripk3 in models of acute and chronic TNF-mediated arthritis and investigate their function in arthritic SFs. The mechanism underlying amelioration of disease in the absence of mesenchymal-specific IKK2 signals involves irregularities in gene expression affecting destructive properties of SFs and differential sensitization of SFs to Ripk1/Ripk3-mediated programmed cell death pathways depending on the chronicity of arthritogenic TNF signals. Our results suggest that complete amelioration of chronic inflammation in arthritic diseases might require a combination of Ripk3 inhibition with mesenchymal targeting of IKK2-mediated signals.

## Results

**Mesenchymal p55TNFR is required for TNF-mediated arthritis.** We have previously shown in TNF-driven models of arthritis that mesenchymal-specific TNF/p55TNFR signalling is sufficient to induce the development of full-blown pathology[6]. To address whether mesenchymal-specific p55TNFR signalling is also required for the development of arthritis, we crossed p55TNFR conditional knockout (KO) mice[29] with $Col6a1$-$Cre$ mice[6] in order to generate animals which lack mesenchymal p55TNFR in their SFs ($p55TNFR^{Ms-KO}$) (Supplementary Figure 1A). $p55TNFR^{Ms-KO}$ animals were crossed into the $hTNFtg$ background and development of disease was monitored. We observed that the $hTNFtg$ p55TNFR-sufficient animals developed full disease at 10 weeks of age, while the $hTNFtg$ $p55TNFR^{Ms-KO}$ animals remained completely healthy (Fig. 1a). Interestingly,

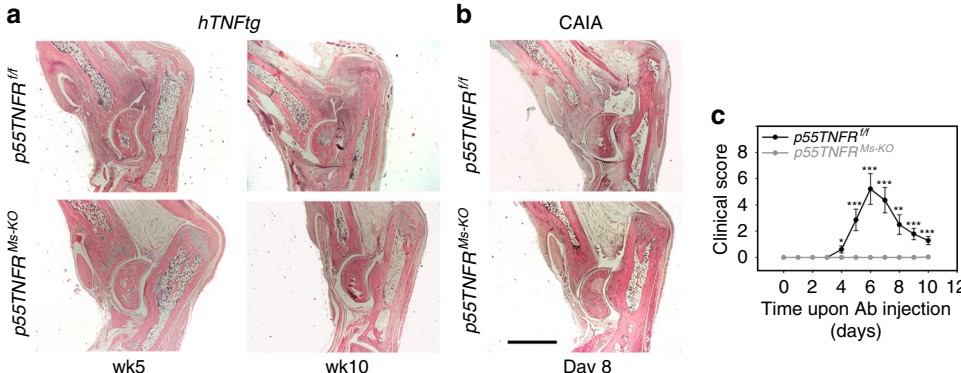

**Fig. 1** Mesenchymal p55TNFR signals are necessary for the development of TNF-mediated arthritis. **a** Representative histological images of haematoxylin & eosin (H&E)-stained ankle joint sections of $hTNFtg$ $p55TNFR^{f/f}$ ($n = 5$, age: week 5; $n = 8$, age: week 10) and $hTNFtg$ $p55TNFR^{Ms-KO}$ mice ($n = 8$, age: weeks 5 and 10). **b** Representative histological images of H&E-stained ankle joint sections of $p55TNFR^{f/f}$ and $p55TNFR^{Ms-KO}$ mice subjected to CAIA at the peak of disease (Day 8). **c** Representative scoring of clinical manifestations of $p55TNFR^{f/f}$ ($n = 7$) and $p55TNFR^{Ms-KO}$ mice ($n = 8$) subjected to CAIA (total 3 experiments, $n = 20$ mice per genotype). Arthritis was induced in male mice by i.v. injection of 4 mg collagen II antibody cocktail (Ab) on Day 0 followed by 75 µg LPS i.p. 3 days later. Scale bar: 1 mm. Data are presented as the mean ± SEM. *$P < 0.05$, **$P < 0.01$ and ***$P < 0.001$ by two-tailed Student's $t$-test

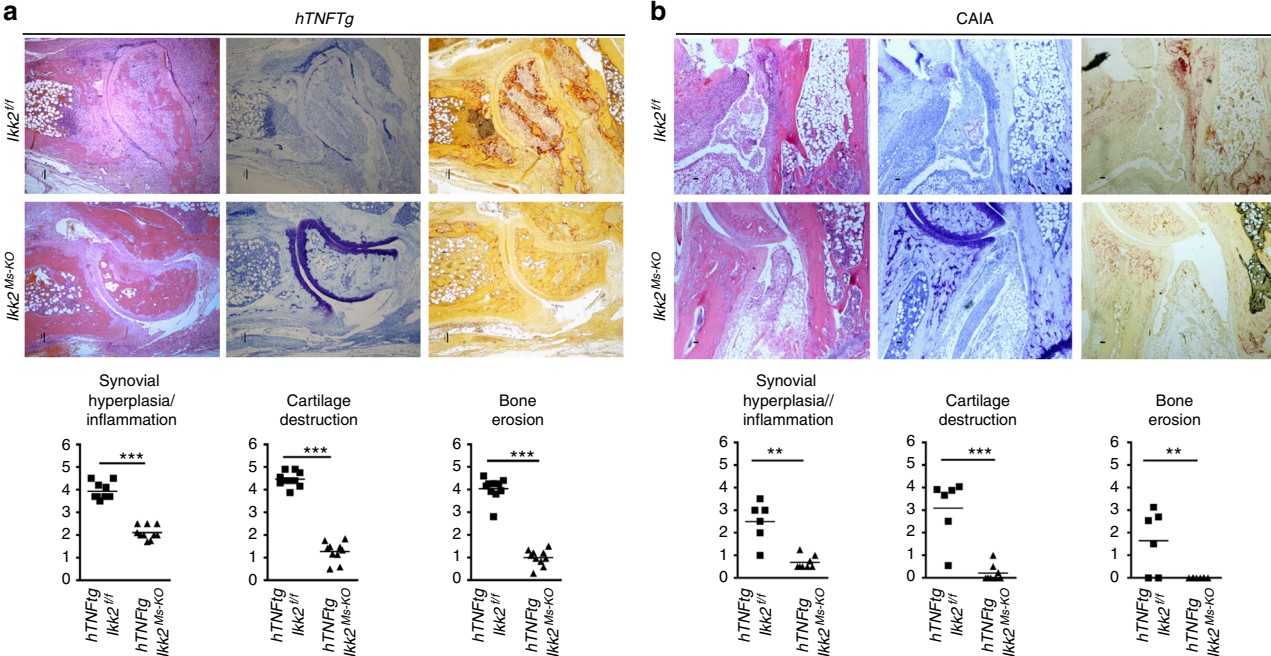

**Fig. 2** Mesenchymal IKK2-dependent signals promote arthritic manifestations in TNF-mediated arthritis. **a** Representative histological images of H&E, Toluidine Blue (TB) and Tartrate-Resistance Acid Phosphatase (TRAP) stained ankle joint sections and histological score of synovial hyperplasia, cartilage destruction and bone erosion of *hTNFtg Ikk2^{f/f}* and *hTNFtg Ikk2^{Ms-KO}* mice (n = 10; age: 11 weeks). **b** Representative histological images of H&E-, TB- and TRAP-stained ankle joint sections and histological score of synovial hyperplasia, cartilage destruction and bone erosion of *Ikk2^{f/f}* (n = 7) and *Ikk2^{Ms-KO}* mice (n = 8) subjected to CAIA at the peak of disease (Day 8). (3 experiments, n = 20–22 mice per genotype). Scale bar: 100 and 50 μm. Data are presented as the mean ± SEM. **P < 0.01 and ***P < 0.001, by two-tailed Student's *t*-test

inactivation of the p55TNFR in SFs was also inhibitory for the development of arthritis in the *TNF^{ΔARE}* model (Supplementary Figure 1B).

Previous studies on cytokine networks operating in the collagen-antibody-induced arthritis (CAIA) demonstrated the essential role of p55TNFR in the development of this induced arthritic pathology[30]. To examine the cell-specific p55TNFR requirements in the CAIA model, we generated *p55TNFR^{Ms-KO}* mice that were injected with the anti-collagen antibody (Ab) cocktail. None of the *p55TNFR^{Ms-KO}* mice developed any joint inflammation either when monitored in the course of the disease or in the histopathological analysis, in sharp contrast to p55TNFR-sufficient littermate controls, which exhibited the reported clinical and histopathological manifestations at the peak of disease (Fig. 1b and c). Thus p55TNFR-mediated signalling in mesenchymal cells of the joints, namely SFs, is required for the development of arthritic disease in three different TNF-dependent models suggesting common and robust arthritogenic mechanisms operating exclusively by p55TNFR on SFs.

**Mesenchymal IKK2 mediates arthritic joint destruction.** Having established a mesenchymal-specific mechanism being required for TNF/p55TNFR mediated arthritogenesis, we investigated the arthritogenic signals acting downstream of p55TNFR in this cell type. NFκB is readily reported in mediating TNF signals in inflammatory settings[7], so we targeted IKK2[31] (encoded by the *Ikbkb* gene and from now on referred as *Ikk2*) in a mesenchymal-specific manner. We generated *Ikk2^{Ms-KO}* animals, which either were crossbred into *hTNFtg* background or subjected to CAIA. Interestingly, the clinical manifestations in both cases were ameliorated (Supplementary Figure 2A and B). Notably, the *hTNFtg Ikk2^{Ms-KO}* animals manifested swelled ankle joints, clinically apparent as early as 3 weeks and throughout the

course of disease (Supplementary Figure 2A) albeit with unaffected grip strength. Detailed histopathological analysis revealed the absence of significant cartilage degradation and bone erosions in the joints of *hTNFtg Ikk2^{Ms-KO}* mice in sharp contrast to the respective findings in their littermate *hTNFtg* IKK2-sufficient mice, even at the advanced stage of the *hTNFtg* arthritic disease (Fig. 2a). Interestingly, synovitis in the *hTNFtg Ikk2^{Ms-KO}* mice was preserved throughout the course of analysis. Similar results were also observed in *Ikk2^{Ms-KO}* animals subjected to CAIA. However, in the collagen-Ab-treated *Ikk2^{Ms-KO}* mice synovitis appeared milder (Fig. 2b). These results show that, in TNF-dependent models of arthritis, the absence of mesenchymal-specific IKK2 is not sufficient to fully attenuate the inflammatory process, while its presence is required to promote the invasive and tissue-destructive properties of arthritogenic SFs.

**IKK2 deficiency leads to synovial death in vivo and ex vivo.** To dissect the mechanisms of mesenchymal-specific IKK2 involvement in the development of arthritis in vivo, we first sought to examine death events in the joint of *Ikk2^{Ms-KO}* mice. We hypothesized that IKK2 deficiency may render *Ikk2*-null SFs sensitive to death as previously described for other cell types[32–34], mainly due to TNF/p55TNFR signals. While the *hTNFtg* and *Ikk2^{f/f}* mice lacked any terminal deoxinucleotidyl transferase-mediated dUTP-fluorescein nick end labelling (TUNEL)-positive cells in the joint area, detection of some cleaved-caspase-3-positive (CC3+) cells, mainly in articular cartilage, was evident in *hTNFtg* mice. The *Ikk2^{Ms-KO}* mice in the arthritic models and, interestingly, also in non-arthritic conditions exhibited a readily detectable signal of TUNEL-positive cells in the all compartments of the synovium (Fig. 3a, b). CC3+ cells were hardly detectable in naive IKK2-sufficient mice; however, in CAIA-subjected mice groups of CC3+ cells were evident in areas of necrotic tissue close

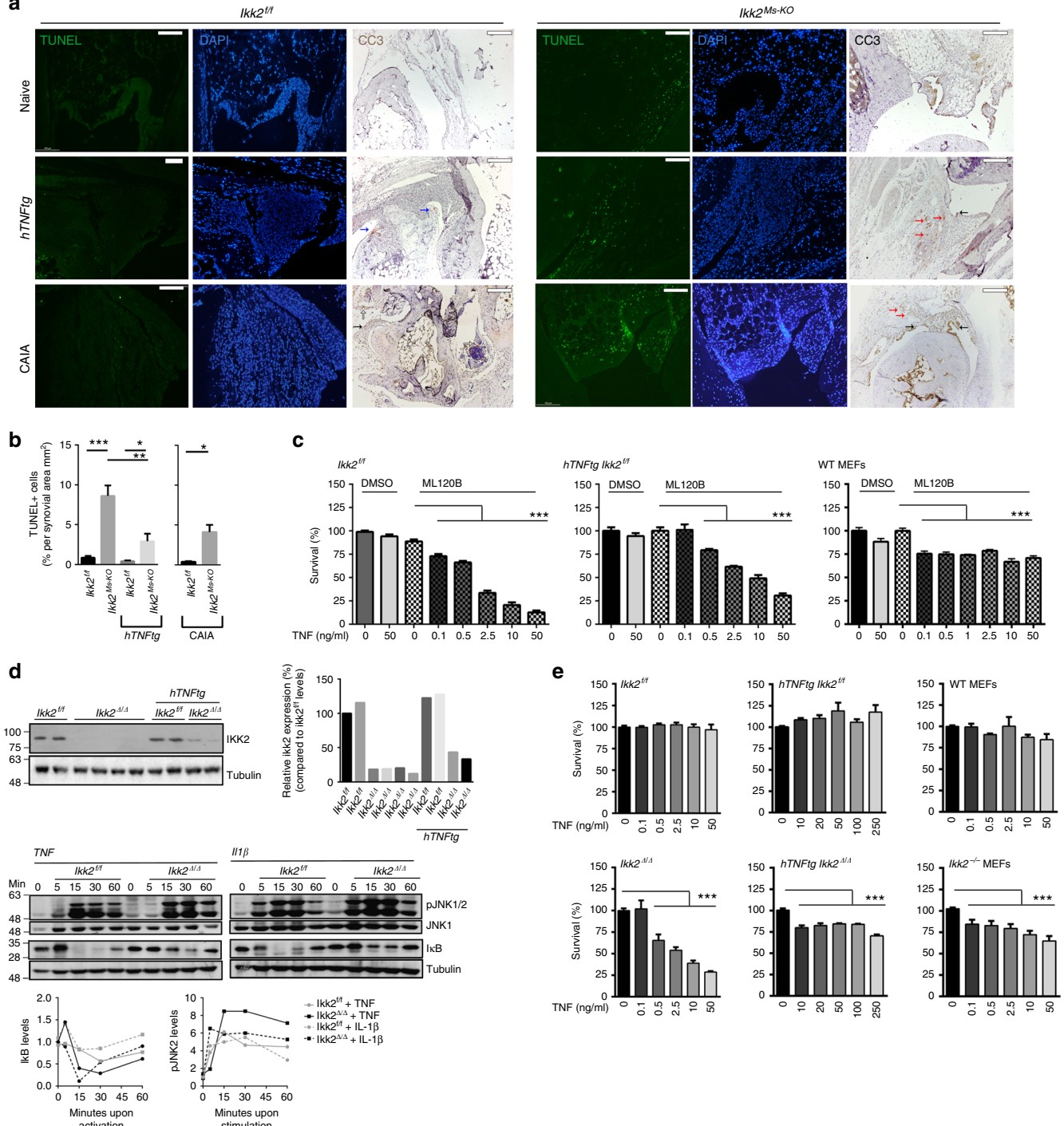

**Fig. 3** Increased death sensitization of Ikk2-deficient SFs in TNF-mediated signals. **a** Representative images depicting TUNEL+ or CC3+ cells in sections of *Ikk2^{f/f}* and *Ikk2^{Ms-KO}* mice (naive, *hTNFtg* (week8) or CAIA-treated (D8)) (*n* = 6 per genotype). Note the CC3+ cartilage cells (blue arrow), lining (black) and sublining (red) SFs, and other cells (open arrow) within necroptic areas. **b** Quantitation of TUNEL+ cells found in synovial membrane area of *Ikk2^{f/f}* and *Ikk2^{Ms-KO}* in naive (*n* = 5–7) and arthritic conditions (*hTNFtg* (*n* = 4–7); CAIA (*n* = 4)). **c** Survival rates of *Ikk2^{f/f}*, *hTNFtg Ikk2^{f/f}* SFs, and WT MEFs treated with different TNF concentrations, in the presence or absence of ML120b inhibitor (representative experiment, *n* = 5). **d** Representative immunoblot of IKK2 expression after the application of TAT-Cre in SF cultures derived from *Ikk2^{f/f}* and *hTNFtg Ikk2^{f/f}* mice, and IkB degradation and JNK1/2 phosphorylation patterns after TNF or IL-1β treatment of *Ikk2*-deficient SFs. Relative quantitations are depicted for each blot. **e** Representative survival rates of *Ikk2^{Δ/Δ}* and *hTNFtg Ikk2^{Δ/Δ}* SFs and *Ikk2^{−/−}* MEFs and control *Ikk2^{f/f}* and *hTNFtg Ikk2^{f/f}* SFs and WT MEFs treated with different TNF concentrations (*n* = 15, 4 experiments). Scale bar: 100 μm. Data are presented as the mean ± SEM. \*P < 0.05, \*\*P < 0.01, and \*\*\*P < 0.001, by two-tailed Student's *t*-test (**b**) or one-way ANOVA (**c**, **d**). Unprocessed original scans of blots are shown in Supplementary Fig. 8

to CC3-negative synovium (Fig. 3a, left panel). In the joints of naive and CAIA-treated $Ikk2^{Ms-KO}$ mice CC3+ cells were evident mainly in synovial lining. However, $hTNFtg\ Ikk2^{Ms-KO}$ mice were characterized by the detection of few CC3+ cells mainly restricted in sublining synovial area. The enrichment of TUNEL+ cells in comparison to CC3+ cells in synovium areas, being mostly obvious in $hTNFtg\ Ikk2^{Ms-KO}$ joints, indicates the ability of IKK2 to prevent more TNF-induced death entities than apoptosis. Additionally, TUNEL positivity in the joint area of $Ikk2^{Ms-KO}$ mice was lost in the absence of p55TNFR (Supplementary Figure 3A), further indicating that p55TNFR/IKK2 signalling is required for the survival of synovial cells in vivo.

To further elucidate the underlying mechanism leading to death events in the joints of $Ikk2^{Ms-KO}$ mice, we established primary cultures of SFs derived from all genotypes. Notably, cultures enriched in IKK2-deficient SFs from either naive or arthritic $Ikk2^{Ms-KO}$ mice could not be achieved (Supplementary Figure 3B) in sharp contrast to the reported high recombination efficiency of $Col6a1$-Cre in combination with other conditional alleles (e.g., Supplementary Figure 1A and ref. [6]). To clarify the potential efficacy problems, we employed $Col6a1$-Cre $Ikk2^{f/f}$ in combination with $Rosa^{mT/mG}$ mice[35], in order to observe Cre-mediated recombination events through the expression of GFP reporter gene. The limited expansion of SFs derived from the double mutant $Col6a1$-Cre $Ikk2^{f/f}\ Rosa^{mT/mG}$ mice, as this was detected by the flow cytometric detection of Cre+ (GFP+) cells in culture, compared to $Col6a1$-Cre $Rosa^{mT/mG}$ SFs suggested defective survival and/or expansion of $Ikk2$-null SFs ex vivo (Supplementary Figure 3C). Furthermore, SFs derived from $Col6a1$-Cre $Ikk2^{f/f}\ p55TNFR^{-/-}$ mice, which additionally lack the p55TNFR signalling and previously shown devoid of TUNEL+ cells in the joint (Supplementary Figure 3A), exhibited successful and functional deletion of IKK2 protein and expanded ex vivo (Supplementary Figure 3B and D), indicating that p55TNFR-mediated signals account for the survival deficits of $Ikk2$-null SFs in ex vivo cultures.

Owing to aforementioned limitations in ex vivo culturing, we first applied an IKK2 inhibitor (ML120B)[19] in WT or $hTNFtg$ SF cultures and then subjected them to TNF-induced death assay (Fig. 3c). In all the IKK2-inhibited cultures tested, even very low TNF doses led to excessive death of SFs (Fig. 3c) being evident as early as 4–5 h upon stimulation. Moreover, when the same conditions (IKK2 inhibitor and TNF) were applied to other cell types, such as mouse embryonic fibroblasts (MEFs; Fig. 3c), none of the cultures exhibited the respective hyper-sensitivity of IKK2-inhibited SFs. We then sought to mimic in vivo conditions and study the function of genetic deficiency of $Ikk2$ in ex vivo SF cultures, a TAT-Cre peptide was applied to $Ikk2^{f/f}$ and $hTNFtg$ $Ikk2^{f/f}$ SF cultures. Both $Ikk2$-null cultures successfully developed and grew ex vivo (Fig. 3d). The ex vivo generated $Ikk2$-null SFs (from now on referred as $Ikk2^{\Delta/\Delta}$) displayed altered TNF- and interleukin (IL)-1-induced responses, as indicated by failure of IkB degradation, whereas phosphorylation of c-Jun N-terminal kinase1/2 (JNK1/2) was prolonged (Fig. 3d) in agreement with published evidence[36]. When different quantities of TNF were applied to $Ikk2^{\Delta/\Delta}$ cultures, we observed hyper-sensitization to death responses similar to the IKK2-inhibited SF cultures (Fig. 3e). Addition of cycloheximide (CHX) in the TNF-treated cultures amplified TNF-induced death indicating the essential role of de novo protein synthesis in survival (Supplementary Figure 3E). The $hTNFtg\ Ikk2^{\Delta/\Delta}$ SF cultures were sensitive to TNF as well (Fig. 3d), yet their sensitivity was lower compared to the IKK2-inhibited $hTNFtg$ SF cultures. Interestingly, inhibition of de novo protein synthesis by CHX sensitized $hTNFtg\ Ikk2^{\Delta/\Delta}$ SFs to death even in the absence of exogenous TNF signals (Supplementary Figure 3E), indicating that a constant influx of

antiapoptotic molecules, such as the CHX-targeted cFLIP$_{(L)}$[37], is required for the prevention of death in $hTNFtg\ Ikk2^{\Delta/\Delta}$ SF cultures. Collectively, our results show that IKK2 signals operating downstream of the p55TNFR in SFs, either acutely or chronically experienced to TNF, are strongly required for their survival in vivo and ex vivo.

**Naive and $hTNFtg$ $Ikk2$-null SFs respond differently to TNF.** We then investigated potential mechanisms that lead to the death of $Ikk2$-deficient SFs in vivo and ex vivo by examining proliferation capacity and the expression of TNF/NFκB-induced genes implicated in cell survival and activation. Synchronized, TAT-Cre-treated SFs of $Ikk2^{f/f}$ and $hTNFtg$ $Ikk2^{f/f}$ cultures (Fig. 4a) were subjected to 3-[4,5-dimethylthiazol-2-yl]-2,5 diphenyl tetrazolium bromide (MTT) incorporation assay revealed defects in overall survival activity of $hTNFtg$ $Ikk2^{\Delta/\Delta}$ compared to $hTNFtg$ $Ikk2^{f/f}$ SF cultures (Fig. 4b). In protein expression analysis level, we identified the NFκB-regulated vascular cell adhesion molecule (Vcam)-1 and intercellular adhesion molecule (Icam)-1 expression of $hTNFtg$ $Ikk2$-null SFs being significantly downregulated compared to the $Ikk2$-null or IKK2–sufficient SFs, which exhibited similar levels of Vcam-1/ Icam-1 expression (Fig. 4b). Surprisingly, hTNF protein expression was almost three-fold upregulated in $hTNFtg$ $Ikk2$-null cell supernatants compared to respective control supernatants from $hTNFtg$ SF cultures (Fig. 4b). We also noticed that, upon TNF stimulation, the caspase3/8-mediated PARP1 cleavage pathway was successfully activated in $Ikk2$-null SFs, indicating the activation of the apoptotic pathway in the absence of NFκB signalling and in sharp contrast to the respective responses of $hTNFtg$ $Ikk2$-null SFs (Fig. 4c, Supplementary Figure 4A). Consistent with the effect of IKK2 deficiency in the activation of caspases upon TNF stimulation, the antiapoptotic molecules cIAP-1, Xiap, and cFLIP$_{(L)}$ as well as the antioxidant enzyme Sod2 failed to upregulate their expression in $Ikk2$-null SFs, whereas this was not evident in the case of $hTNFtg$ $Ikk2$-null SFs (Fig. 4c, Supplementary Figure 4A).

Quantitative PCR analysis revealed the defective RNA expression of NFκB-regulated molecules, such as IL6, RankL (encoded by $Tnfsf11$ gene), $Mmp-3$, -13 and -14 in all $Ikk2$-null SF extracts; however, the $Timp1$ expression remained unaltered (Fig. 4d). The $hTNFtg$ $Ikk2$-null SFs presented a partly attenuated phenotype compared to $hTNFtg$ IKK2-sufficient SFs, characterized by spontaneous cytokine and matrix metalloproteinase (MMP) gene expression, which is merely changed upon further TNF stimulation (Fig. 4d). Interestingly, naive $Ikk2$-null cells exposed to TNF presented slightly upregulated the $Il1b$ RNA expression; however, the difference did not reach statistical significance compared to TNF-exposed IKK2-expressing SFs (Fig. 4d). Collectively, these results indicate that, depending on the chronicity of TNF signals imposed on SFs, proinflammatory and survival gene expression responds differentially to the absence of IKK2 signals, with acute TNF signals (such as transiently induced in CAIA conditions) leading $Ikk2$-deficient SFs mainly to changes in proinflammatory gene expression and apoptotic activation, while in contrast, chronic TNF-experienced $Ikk2$-null SFs ($hTNFtg$ background) exhibited less profound changes in pro-inflammatory activation and no overt differences in antiapoptotic responses.

**IKK2 abrogates TNF-induced Ripk1-dependent death in SFs.** To gain insight into the mechanisms that regulate the differential apoptotic response of $Ikk2$-deficient SFs and the sensitivity of the $hTNFtg$ $Ikk2$-deficient SFs to a caspase3/8-independent death, we subjected the $Ikk2$-null SF cultures to TNF treatment in the presence of pan-caspase and/or Ripk1/Mlkl inhibitors (zVAD or

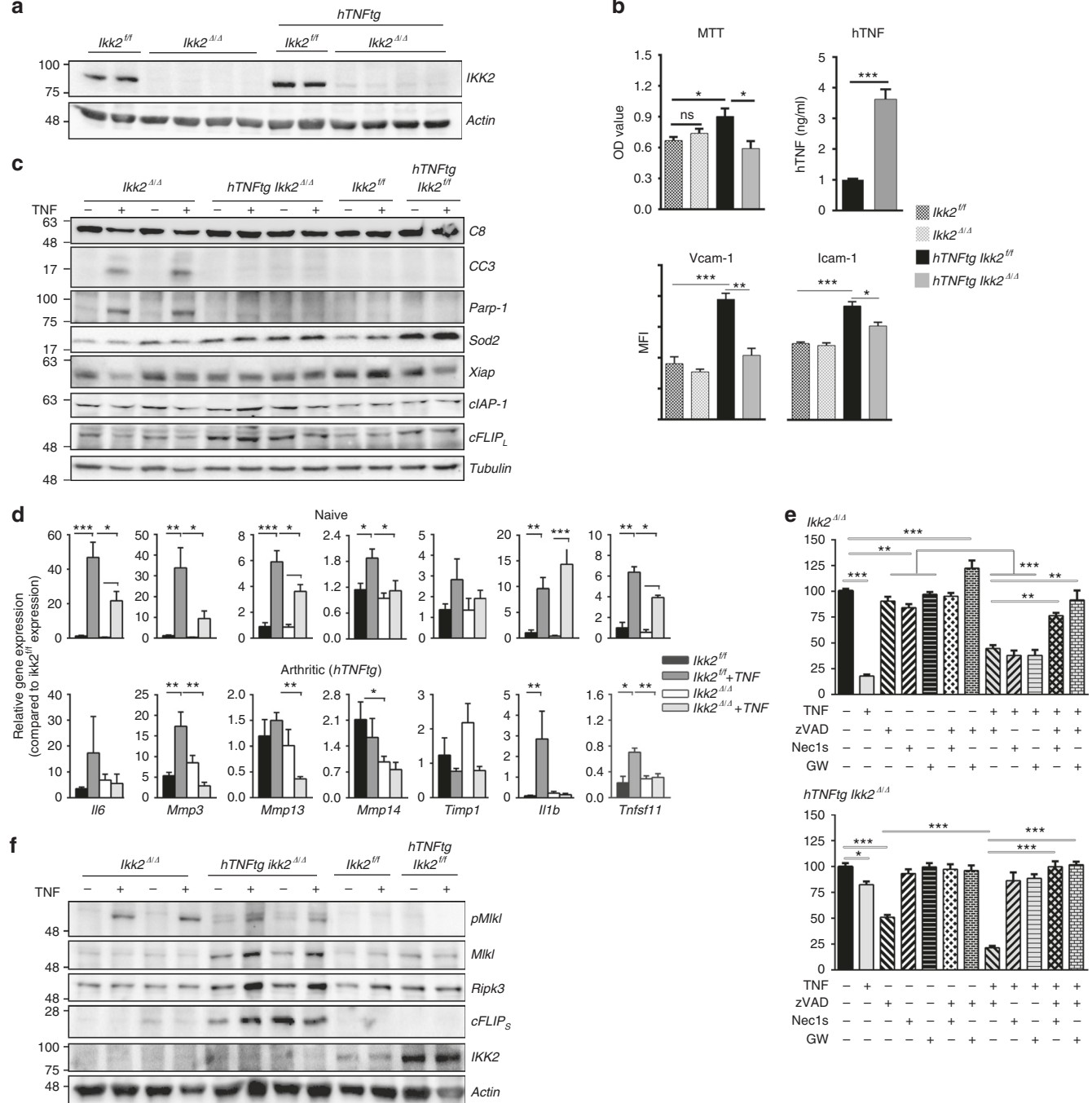

**Fig. 4** IKK2 deficiency differentially affects gene expression of naive and arthritic SFs. **a** Representative immunodetection of IKK2 levels in TAT-Cre treated *Ikk2^f/f* (*Ikk2^Δ/Δ*) and TAT-Cre treated *hTNFtg Ikk2^f/f* (*hTNFtg Ikk2^Δ/Δ*) SFs compared to non-treated control cultures *Ikk2^f/f* and *hTNFtg Ikk2^f/f*, respectively (upper panel). **b** Representative MTT incorporation assay in the indicated genotypes; hTNF quantitation in the supernatants of *hTNFtg Ikk2^Δ/Δ* (grey column) and control *hTNFtg Ikk2^f/f* SF cultures (black column) (*n* = 7, 2 experiments); flow cytometric detection of Vcam-1 and Icam-1 levels in cultured *hTNFtg Ikk2^f/f* (black), *hTNFtg Ikk2^Δ/Δ* (grey) and control SFs (*Ikk2^f/f*: dotted black and *Ikk2^Δ/Δ*: dotted grey) (*n* = 4, 2 experiments). **c** Representative immunodetection of Casp8 (C8), Cleaved Casp3 (CC3), Parp1, Sod2, Xiap, cIAP-1 and cFLIP(L) in cell extracts of *Ikk2^Δ/Δ*, *hTNFtg Ikk2^Δ/Δ* and control SF cultures treated with TNF (20 ng/ml, 5 h) (*n* = 10). **d** Relative gene expression of the NFκB-regulated genes *Il6*, *Mmp-3*, *-13*, *-14*, *Timp1*, *Il1b* and *Tnfsf11* (RankL) (normalized to b2m levels) in SFs of the indicated genotypes upon TNF (20 ng/ml, 5 h) (*n* = 3–8). **e** Survival rates of *Ikk2^Δ/Δ* and *hTNFtg Ikk2^Δ/Δ* SFs treated with TNF, zVAD, Nec1s and GW806742X (GW) as indicated (*n* = 3). **f** Representative immunodetection of pMlkl, Mlkl, Ripk3, cFLIP(S) and IKK2 levels in TNF-treated *Ikk2^Δ/Δ*, *hTNFtg Ikk2^Δ/Δ* and control SF cultures (TNF: 20 ng/ml, 5 h) (*n* = 8, 4 experiments). Data are presented as the mean ± SEM. *P < 0.05, **P < 0.01 and ***P < 0.001, by two-tailed Student's *t*-test (**b**, **d**) or one-way ANOVA (**e**). Unprocessed original scans of blots are shown in Supplementary Fig. 8

Nec1s/GW806742X inhibitor[38], respectively). Interestingly, TNF-induced death was partially rescued in *Ikk2*-null SFs by either treatment, indicating that besides the caspase-mediated death, a caspase-independent and Ripk1/Mlkl-mediated death is also at play. The *hTNFtg Ikk2*-null SFs, even in the absence of additional TNF treatment, were extremely sensitive to caspase inhibition that led to a high percentage of dead cells (Fig. 4e). This zVAD-induced death was prevented by co-treatment with either Nec-1s or Nec-1 (data not shown; similar effects to Nec1s) or GW806742X inhibitor indicating a zVAD-induced, Ripk1/Mlkl-dependent death taking place in *hTNFtg Ikk2*-null SFs (Fig. 4e). To explore the displayed sensitivity of *hTNFtg Ikk2*-null SFs to necroptotic-like events, we followed the Mlkl phosphorylation upon TNF stimulation, a key event in necroptotic death. We identified Mlkl phosphorylation in *Ikk2*-null SFs upon a 5 h TNF stimulation (Fig. 4f) even in the presence of effective caspase activation (Fig. 4c), and low but readily detectable and spontaneous Mlkl phosphorylation happening in the *hTNFtg Ikk2^{Δ/Δ}* SFs, which was further enhanced in the presence of TNF (Fig. 4f, Supplementary Figure 4B). We also noticed the upregulation of both Ripk3 and Mlkl levels in the *hTNFtg Ikk2^{Δ/Δ}* cells, indicating that these upregulations may account for the spontaneous Mlkl phosphorylation events (Fig. 4f). Notably, the expression of FLIP(s), a form of FLIP implicated in the predisposition of cells to necroptosis through its regulatory function in the ripoptosome complex[39], was upregulated in the *hTNFtg Ikk2^{Δ/Δ}* cultures (Fig. 4f) further implicating a necroptotic pathway being active in the *hTNFtg Ikk2*-null cells. Collectively, these data indicate an unusual potential of TNF-stimulated *Ikk2*-deficient SFs to program their cell death through both caspase-dependent and -independent pathways, with the balance tipping towards to Ripk1-mediated caspase-independent death when these cells have been chronically exposed to TNF signals.

**Ripk3 ablation abrogates synovitis in *hTNFtg Ikk2^{Ms-KO}* mice.** Following our ex vivo evidence on the susceptibility of *hTNFtg Ikk2*-null SFs to different death events accompanied by an upregulation of Ripk3 levels, we investigated whether the altered gene expression and the death predisposition could account for the persisting synovitis manifested in *hTNFtg Ikk2^{Ms-KO}* mice. Ripk3 acts downstream of Ripk1 in the TNF pathway, regulating both cellular demise and gene expression (reviewed in ref. [40]). The absence of any effect of Ripk3 deficiency in the development of arthritis in *hTNFtg* mice (Supplementary Figure 5A, B) indicated that Ripk3-mediated signals are dispensable for the arthritogenic behaviour of *hTNFTg* SFs in vivo. Interestingly when we generated *hTNFtg Ikk2^{Ms-KO}* mice crossbred into *Ripk3^{−/−}* background, we observed that mice did not develop any sign of arthritic disease. Histological analysis of the ankle joints supported the results of the clinical examination, revealing intact joint architecture and absence of any signs of synovitis, cartilage destruction and bone erosions compared to *hTNFtg Ikk2^{Ms-KO}* mice (Fig. 5a and Supplementary Figure 6A). Histomorphometric analysis in the cancellous bone tissue of calcaneus, a bone primarily affected in *hTNFtg* arthritis, confirmed that several bone indices were comparable to the non-arthritic littermate controls (Supplementary Figure 6B).

To dissect the molecular aspects of the Ripk3 pathway in the behaviour of *Ikk2*-null SFs, we generated *Ikk2/Ripk3*-null cultures ex vivo from both naive and *hTNFtg* mice (Supplementary Figure 6C). Ripk3 deficiency did not affect the elevated levels of Vcam-1 and Icam-1 on the cell surface of *hTNFtg* SFs. Moreover, it did not alter the already downregulated expression of these proteins in *hTNFtg Ikk2^{Δ/Δ} Ripk3^{−/−}* SFs (Supplementary Figure 6D) compared to *hTNFtg Ikk2^{Δ/Δ}* SFs and in sharp

contrast to *hTNFtg* IKK2-sufficient controls (Fig. 4b). Remarkably, even though the single Ripk3 deficiency did not alter the hTNF expression levels of *hTNFtg* SFs, however, the high hTNF levels detected in *hTNFtg Ikk2^{Δ/Δ}* supernatants were normalized in the concomitant absence of IKK2 and Ripk3, reaching the respective levels of *hTNFtg* IKK2-sufficient controls (Supplementary Figure 6E) indicating that, only in the absence of IKK2, Ripk3-mediated signals regulate hTNF expression.

The SF cultures were also subjected to TNF/zVAD/Nec1 treatments. The *hTNFtg Ikk2/Ripk3*–null SFs were completely protected by TNF- and zVAD-induced death confirming ex vivo that Ripk3-mediated signals are responsible for the cell death seen in *hTNFtg Ikk2^{Δ/Δ}* SFs (Fig. 5b). TNF treatment of control *Ikk2^{Δ/Δ} Ripk3^{−/−}* SFs led to a similar pattern of death compared to *Ikk2^{Δ/Δ}* SFs (Figs. 5b and 3d, respectively). All TNF/zVAD-treated *Ikk2/Ripk3*-null cultures tested retained their ability to fully survive as the non-treated control in contrast to the partial death provoked by activation of caspases in the TNF/Nec1-treated *Ikk2/Ripk3*-null SFs (Fig. 5b), indicating that Ripk3 deficiency does not alter the apoptotic responses previously observed in *Ikk2*-deficient SFs. Consistent with the effects of Ripk3 deficiency on the TNF-induced death assays, the Mlkl phosphorylation was absent from any *Ikk2/Ripk3*-null culture (Fig. 5c), whereas caspase-3/8 activation, as well as XIAP expression followed the expected patterns caused by *Ikk2* deficiency upon TNF stimulation (Fig. 5d). Interestingly, the *hTNFtg Ikk2^{Δ/Δ} Ripk3^{−/−}* SFs presented a slight activation of caspases; however, FLIP(S) expression remained elevated compared to *Ikk2^{Δ/Δ} Ripk3^{−/−}* cells (Fig. 5d).

Collectively, these findings exclude a direct role of Ripk3 involvement in TNF-mediated arthritis. Moreover, our evidence indicates that the inhibition of Ripk3 would act beneficially only in the absence of mesenchymal IKK2-mediated signals in chronic arthritis. Mechanistically, the Ripk3 pathway conditionally controls gene expression and/or the cell death decisions, which were observed ex vivo in the *hTNFtg Ikk2*-deficient SFs thus suggesting that the persisting synovitis in *hTNFtg Ikk2^{Ms-KO}* animals could be provoked by a Ripk3-mediated inflammatory process.

**TNF induces RIPK1-mediated death in proteasome-inhibited RA-SFs.** Previous studies involving genetic or pharmacological inhibition of IKK2 in human RA-SFs suggested a role of IKK2-mediated signalling affecting mainly proinflammatory gene expression in the absence of death responses[19,22,23]. These results indicated that IKK2 is not a prominent NFκB inhibitor that affects pro-survival signalling in RA-SFs. Here we examined the expression levels of necroptosis related molecules RIPK3 and MLKL. We noticed that both molecules are expressed in RA-SFs and osteoarthritis (OA)-SFs with no significant differences between the two types (Supplementary Figure 7A, B). We then sought to examine an experimental setting that affects survival of human SFs by affecting NFκB responses. MG132 effectively blocks the proteolytic activity of the 26S proteasome complex thus affecting pathways including NFκB and inducing apoptotic cell death[41]. Importantly, it was shown to affect survival of human SFs too[42]. Thus, upon MG132/TNF treatments, both RA- and OA-SFs from five patients were prone to death (Supplementary Figure 7C). Individual treatments with either caspase inhibitor or Nec1 marginally (but not statistically significant) rescued cells from death, abrogating any favouring for any of the two death pathways in any cell type. In sharp contrast, the combinational treatment with both inhibitors fully restored survival of TNF/MG132-treated SFs (Supplementary Figure 7C). Interestingly, by looking at individual sample responses

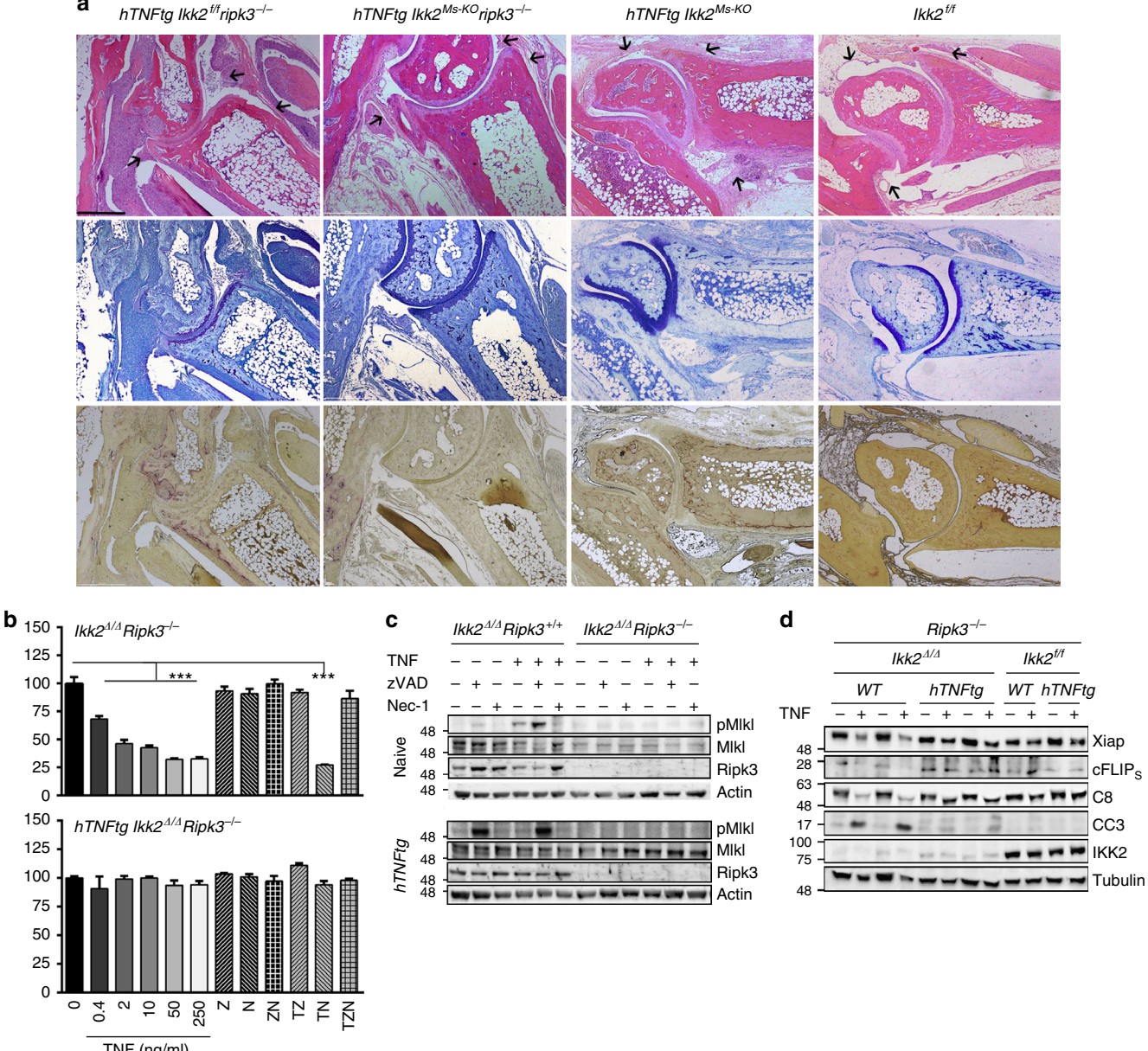

**Fig. 5** Ripk3-mediated signals are indispensable for persisting synovitis in hTNFtg *Ikk2*^Ms-KO mice. **a** Representative histological images of H&E-, TB- and TRAP-stained ankle joint sections of 8-week-old *hTNFtg Ikk2^f/f Ripk3^−/−* (n = 11) and *hTNFtg Ikk2^Ms-KO Ripk3^−/−* mice (n = 13) in comparison to *hTNFtg Ikk2^Ms-KO* and *Ikk2^f/f* mice. Note the differences in synovium and adjacent tissues in respective areas indicated by the arrows. **b** Survival rates of *Ikk2^Δ/Δ Ripk3^−/−* and *hTNFtg Ikk2^Δ/Δ Ripk3^−/−* SF cultures treated with TNF (T), zVAD (Z), and nec-1 (N) as indicated. **c** Representative immunodetection of pMlkl, Mlkl and Ripk3 levels in the cell extracts of *Ikk2^Δ/Δ Ripk3^−/−* and *hTNFtg Ikk2^Δ/Δ Ripk3^−/−* SFs treated as indicated compared to control *Ikk2^Δ/Δ Ripk3^+/+* and *hTNFtg Ikk2^Δ/Δ Ripk3^+/+* samples, respectively (n = 4 per genotype, 2 experiments). **d** Representative immunodetection of Casp8 (C8), Cleaved Casp3 (CC3), Xiap, cFLIP(s) and IKK2 levels in cell extracts of *Ikk2^Δ/Δ Ripk3^−/−* and *hTNFtg Ikk2^Δ/Δ Ripk3^−/−* SFs treated as indicated compared to control *Ikk2^f/f Ripk3^−/−* and *hTNFtg Ikk2^f/f Ripk3^−/−* samples, respectively (the non-hTNFtg control cultures are indicated as WT) (n = 4 per genotype, 2 experiments). Scale bar: 500 μm. Data are presented as the mean ± SEM. ***P < 0.001 one-way ANOVA test (**b**). Unprocessed original scans of blots are shown in Supplementary Fig. 8

(Supplementary Figure 7D), we noticed that 2 out of 5 samples (40%) from both OA and RA patients have been partially rescued by each inhibitor treatment suggesting that the TNF/MG132-mediated effects are not limited to a caspase-mediated death but also includes another RIPK1-mediated cellular demise in the SFs of some patients. We also confirmed the RIPK1 specificity by applying Nec1s inhibitor in responsive samples, which indeed showed a similar pattern of responses (Supplementary Figure 7E). These results indicate the involvement of differential death responses in the absence of intact survival signalling in human SFs as we noticed in murine *Ikk2*-null SFs.

## Discussion

We have previously demonstrated the sufficiency of TNF/p55TNFR-signalling in SFs for the orchestration of murine TNF-dependent arthritis[6]; however, the requirement of this cell-specific pathway for the development of disease has remained

unclear. In this study, by employing the arthritic *hTNFtg* and *TNF^ΔARE* models in combination with cell-specific inactivation of the p55TNFR, we provide novel evidence for the indispensable role of mesenchymal-specific TNF/p55TNFR signalling in the pathogenesis of modelled TNF-mediated arthritis. Furthermore, we demonstrate the key function of this pathway across models by showing its requirement also in the CAIA model[43,44]. We have thus unraveled a common cellular basis explaining TNF-mediated inflammatory and destructive arthritis, independently of whether stromal (as in *hTNFtg*)[6], innate (as in *TNF^ΔARE*)[6] or adaptive (as in CAIA)[43] stimuli are instigating pathology. These findings may translate into respective TNF-driven mechanisms operating in several subclasses of human arthritides and explain the comparatively broad success of anti-TNF therapies in RA patients.

In light of this evidence, we then examined the mesenchymal-specific function of IKK2, a natural mediator of TNF/p55TNFR signals. IKK2 deficiency in mice is lethal due to a massive apoptotic response in the embryonic liver caused by defective NFκB activation[32–34]. Cell-specific KO approaches revealed the essential role of IKK2 in TNF- and TLR-mediated activation of NFκB and its importance in regulating inflammatory and proliferative as well as death responses (reviewed in ref. [45]). Owing to such pro-inflammatory and pro-survival functions, NFκB appeared as an appealing pharmacological target. Indeed, several IKK2 inhibitors have been developed; however, almost none achieved the criteria to pass into clinical applications. This was mainly due to the lack of specificity and multiple side effects that were poorly reported in the preclinical studies most probably due to the short term application of the protocols (reviewed in ref. [46]). Interestingly, longer IKK2 inhibition led to leukocyte death and increased neutrophilia, rendering the mice severely immunocompromised[47], susceptible to endotoxemia[48] and autoimmune diseases[49], indicating that inactive IKK2 leads to miscellaneous responses depending on the cell target. Our observations revealed that targeted mesenchymal IKK2 deficiency did not lead to overt phenotypes. By employing the *hTNFtg* model of chronic arthritis, we observed that, regardless of the ameliorated arthritic phenotype and the preserved joint architecture, mice with joint mesenchymal cells deficient in IKK2 still developed a sustained synovitis throughout the course of chronic disease, indicating that IKK2 acts as a checkpoint regulator of the progression from synovitis to pannus formation.

Mechanistic ex vivo studies on *Ikk2*-null SFs suggest that the naive *Ikk2*-null SFs exhibit an inflammatory and antiapoptotic transcriptional insufficiency caused by the defective NFκB and caspase-3 activation in response to TNF, in line with previous reports[50]. Surprisingly, the *hTNFtg Ikk2*-null SF population was selectively expressing some inflammatory profiles despite their chronic TNF exposure (Fig. 4b, d). However, their resistance to activate caspase-3 ex vivo could be correlated with unaltered antiapoptotic responses, such as IAPs (Fig. 4c), similar to what has been reported for RA-SFs[51].

Biochemical approaches employing *Ikk2*-null SFs revealed the extreme sensitivity of cells to death in response to even minute concentrations of TNF compared to the more resistant *Ikk2*-null MEFs, suggesting that the threshold of responsiveness of SFs to TNF may be much lower compared to other mesenchymal cell types. The application of a pan-caspase inhibitor, a specific Ripk1 inhibitor and an Mlkl inhibitor in both naive and *hTNFtg Ikk2*-deficient SFs revealed that, besides the expected caspase-dependent death (apoptosis), a caspase-independent, yet Ripk1/Mlkl-dependent death is taking place in the absence of IKK2. Even though the GW806742X inhibitor (targeting Mlkl) had been recently reported to affect the human RIPK1/3 activity too[52], the TNF-induced phosphorylation of Mlkl in *Ikk2*-null SFs and the spontaneous phospho-Mlkl expression in *hTNFtg Ikk2^Δ/Δ* SFs

(chronically exposed to TNF) indicate a possible Mlkl involvement in our experimental system. Moreover, we show that IKK2 regulates hTNF, Ripk3 and Mlkl protein expression under chronic inflammatory conditions (*hTNFtg Ikk2^Δ/Δ* SFs; Fig. 4b, f). In turn, changes in the FLIP$_L$/FLIP$_S$ ratio may prevent necrotic events caused by the insufficient caspase-8 release from apoptosome, favouring necrosome rather than apoptosome function as previously reported[39,53,54]. These ex vivo results are well correlated with the in vivo detection of death events marked by TUNEL reactivity in the joint mesenchyme in sharp contrast to the control IKK2-sufficient mice, which developed full-blown arthritis. Further analysis employing cleaved caspase-3 detection in vivo stratified the death events according to the genotype, the chronicity of inflammatory conditions and, interestingly, the anatomic location; even though TUNEL indiscriminately marked the *Ikk2*-deficient synovium area irrespectively of the naive or arthritic conditions, CC3+ cells were mainly located in the lining part in naive and CAIA-treated mice (acute arthritis). Notably, the *hTNFtg Ikk2*-deficient synovium is surprisingly less prone to CC3-mediated apoptotic events marked with a few CC3+ cells, located mainly in sublining area regardless of uniform staining of TUNEL throughout the synovium. This and our ex vivo findings suggest that either Mlkl phosphorylation and caspase activation is happening within the same *Ikk2*-deficient cell or that distinct cells within the *Ikk2*-deficient SF population die by different modes of death. In this regard, it is possible that distinct SF types exist within the synovium and differentially regulate local responses that affect the physiology of the joints leading to pathologic situations. Moreover, numerous reports have analysed the resistance of arthritic SFs to apoptosis (reviewed in ref. [51]) rendering the re-activation of their death machinery as a potential therapeutic approach. However, the segregation of programmed cell death into apoptosis, necroptosis, pyroptosis, etc, depending on the molecular players and the processes involved, has started clarifying their functional significance in physiology and disease. The inflammatory nature of necroptotic death due to the release of damage-associated molecular patterns compared to the immunological quiescent apoptotic death[40] may explain the abrogation of the residual inflammation seen in the synovium in the absence of both IKK2 and Ripk3 signalling in SFs, whereas Ripk3 were dispensable for the development and progression of the chronic TNF-mediated arthritis. Interestingly, this latter finding contrasts the previously reported acceleration of resolution phase of arthritis caused by the Ripk3 deficiency in the KRN-serum transfer arthritis model[13]. Of note, the KRN-serum transfer model of arthritis is minimally TNF dependent and more TLR4 and IL-1β dependent[55], and the diminished TLR4/TRIF/Ripk3-dependent IL-1β expression by macrophages, and not macrophage necroptosis, could explain the observed differences[13]. In our study, we clearly show that the mesenchymal compartment of the *hTNFtg* joints, namely SFs, is the major responder to TNF arthritogenic signals that could act independently to induce synovitis even in defective IL-1 conditions[56,57], as this could happen, e.g., in a *Ripk3*-null background. Moreover, IKK2/Ripk3 deficiency is rather relevant to the downregulation of hTNF release by the double IKK2/Ripk3-deficient *hTNFtg* SFs compared to *hTNFtg Ikk2*-null SFs (Suppl. Figure 6E). The double deficiency could be also associated with the loss of spontaneous Mlkl phosphorylation and the diminished caspase-independent death seen ex vivo as well as the neutralization of arthritis seen in vivo.

Another possible explanation for the differential death susceptibility of *Ikk2*-deficient SFs could be attributed to the phosphorylation state of Ripk1, as it was recently exhibited that the hypophosphorylated Ripk1 caused by compound IKK1/2 deficiency is responsible for the detection of different types of cell

death (caspase-3/8-mediated apoptosis or Ripk3-mediated necroptosis) depending on the context and independently of NFκB activation; however, single *Ikk2* deficiency was not sufficient to provide the same result in that study[58]. Interestingly, another recent report suggested non-necroptotic functions of either Mlkl and/or Ripk3 in controlling, at least partially, lymphoid tissue homeostasis via production of chemokines and cytokines under caspase-8 or FADD deficiency[59]. Whether Ripk3 could further regulate altered gene expression in the arthritic *Ikk2*-null SFs, as it has been reported for other cell types deficient in IAPs[13,60–63], and/or death is yet to be explored. Further studies employing several mutant mice for the molecular players of necroptotic and apoptotic pathway will shed light on the contribution of death pathways under mesenchymal IKK2 deficiency in TNF-mediated arthritis.

Interestingly, even though IKK2 inhibition in OA- or RA-SFs is not sufficient to induce similar effects[19], MG132, a potent NFκB modulator acting through proteosomal inhibition and suggested for the treatment of arthritis reviewed in[64], provoked both RIPK1-mediated caspase-dependent and -independent death in SFs from some patients, providing translational evidence to our murine-based observations.

In conclusion, our study provides novel mechanistic insights into the mesenchymal-specific role of the IKK2 and Ripk3 axes during TNF-dependent arthritogenesis, offering ex vivo and in vivo evidence on their complementary functions in regulating differential modes of pro-inflammatory synovial cell activation and death with significant impact on the physiological outcome in modelled arthritis. Moreover, the distinct property of *Ikk2*-null SFs to switch from a caspase-dependent to a Ripk3-mediated caspase-independent mode of death following chronic TNF stimulation suggests that combination of mesenchymal IKK2 targeting along with Ripk3 blockade may be considered as an optimal therapeutic strategy for the treatment of chronic arthritis.

## Methods

**Mice and induction of arthritis.** Human TNF transgenic (*hTNFtg*), *TNF*^ΔARE, *Col6a1-Cre*[6], *p55TNFR*^{f/f}[29], *Ikk2*^{f/f}[33], *p55TNFR*^{−/−}[65], *Rosa*^{mT/mG}[37] and *Ripk3*^{−/−}[66] mice have been previously described. *Col6a1-Cre Ikk2*^{f/f} and *Col6a1-Cre p55TNFR*^{f/f} mice are referred to as *Ikk2*^{Ms-KO} and *p55TNFR*^{Ms-KO}, respectively. All mice were maintained on a C57BL/6J except from hTNFtg *Ikk2*^{Ms-KO} and littermate mice, which were maintained in a CBA;C57BL/6J genetic background. Mice were maintained under specific pathogen-free conditions in conventional, temperature-controlled, air-conditioned animal house facilities of BSRC Alexander Fleming with 14–10 h light/dark cycle and received food and water ad libitum.

CAIA was induced with a cocktail of four monoclonal IgG Abs, specific for four well-defined major epitopes on triple helical Collagen II, the C1 (CIIC1 clone), J1 (M2139 clone), D3 (CIIC2 clone) and U1 epitopes (UL1 clone)[67] (MD Biosciences; ArthritoMab^TM; #CIA-MAB-2C). Briefly, arthritis was induced in 6–8-week-old littermate male mice of the indicated genotypes by intravenous injection of ArthritoMab^TM (4 mg) on day 0. On day 3, mice also received an intraperitoneal injection of 75 µg lipopolysaccharide diluted in 150 µl phosphate-buffered saline (PBS; *E. coli* strain 0111B4; MD Biosciences; #CIA-MAB-2C).

**Clinical assessment of arthritis.** Arthritis in *hTNFtg* animals was evaluated in ankle joints in a blinded manner using a semiquantitative arthritis score ranging from 0 to 4: 0: no arthritis (normal appearance and grip strength); 1: mild arthritis (joint swelling); 2: moderate joint swelling and digit deformation, reduced grip strength; 3: severe joint swelling and digit deformation, impaired movement, no grip strength; 4: severe joint swelling and digit deformation, impaired movement, no grip strength and obvious weight loss cachexia.

CAIA clinical evaluation was performed as previously described[68]. Briefly, arthritis was evaluated in a blinded manner for each of hindpaw and forepaw, according to the following scale: 0 for normal appearance, 1 for swelling and/or redness in one joint, 2 for swelling and/or redness in more than one joint, 3 for swelling and/or redness in the entire paw, and 4 for maximal swelling. The final score value is the sum of the scores obtained from the four limbs of each mouse.

**Histology.** Formalin-fixed, EDTA-decalcified, paraffin-embedded mouse tissue specimens were sectioned and stained with haematoxylin and eosin (H&E), Toluidine Blue or Tartrate-Resistance Acid Phosphatase (TRAP) Kit [Sigma-

Aldrich]. H&E-, TB- and TRAP-stained joint sections were semiquantitatively blindly evaluated for the following parameters: synovial inflammation/hyperplasia (scale of 0–5), cartilage erosion (scale of 0–5), and bone loss (scale of 0–5) based on an adjusted, previously described method[69] (Supplementary Table 1).

**Immunohistochemistry and TUNEL.** TUNEL was performed in tissue sections according to manufacturer's instructions [DeadEnd Fluorometric Tunel system; Promega: #G3250]. The slides were deparaffinized, rehydrated and permeabilized with 20 µg/ml proteinase K for 25 min at room temperature. The TdT-mediated labelling reaction was performed for 60 min at 37 °C using fluorescent-labeled nucleotides. DAPI [4,6-diamidino-2-phenylindole; Santa Cruz Biotechnology, Inc.] was used to stain the nuclei. Quantitation of TUNEL+ cells were performed with the ImageJ software.

Cleaved caspase-3 detection was performed according to Ab manufacturer's instructions [Cell Signalling; #9661; 1:200]. Briefly, the slides were deparaffinized and rehydrated. Antigen unmasking was performed in 1× Citrate Buffer, 45 min, at 90 °C. Upon endogenous peroxidase quenching and blocking, the incubation with the Ab was performed overnight, and visualization was performed using the ABC and DAB Kit [Vector laboratories; #PK-6100 and #SK-4105, respectively].

**Microcomputed tomography.** Microcomputed tomography (mCT) of excised joints was carried out by a SkyScan 1172 CT scanner (Bruker, Aartselaar, Belgium) following the general guidelines used for assessment of bone microarchitecture in rodents using mCT[70]. Briefly, scanning was conducted at 50 kV, 100 mA using a 0.5-mm aluminum filter, at a resolution of 6 mm/pixel. Reconstruction of sections was achieved using the NRECON software (Bruker) with beam hardening correction set to 40%. The analysis was performed on a spherical volume of interest (diameter 0.53 mm) within 62 slides of the trabecular region of calcaneus. Morphometric quantification of trabecular bone indices such as trabecular bone volume fraction (BV/TV%), bone surface density (BS/TV%), trabecular number (Tb. N; 1/mm) and trabecular separation (Tb. Sp; mm) were performed using the CT analyser program (Bruker).

**Cell culture.** WT and *Ikk2*^{−/−} MEFs were previously described[71]. Primary mouse SFs were isolated from mice with the indicated genotypes and cultured for four passages as previously described[72]. For Cre transduction experiments, $5 \times 10^5$ cells/well (from passage 2) were plated on a 6-well plate, washed with PBS and incubated for 16 h in 1:1 filtered mixture of protein-free Dulbecco's modified Eagle's medium (DMEM) and PBS containing 3 µM TAT-Cre (Millipore; #SRC508). After transduction, cells were washed and cultured at least 2 days in normal DMEM/10% foetal bovine serum, and transduced cultures were further expanded for 3 more passages.

In experiments with the IKK2 inhibitor, pretreatment of cells with ML120B (10µM)[19] was performed for 3 h before addition of other reagents. zVAD [Bachem; #N-1560], Nec-1 [Sigma-Aldrich; #N9037], Nec1s [BioVision; #2263] and GW806742X [SYNkinase; #SYN-1215] were used in concentrations of 20, 30, 30 and 0.5 µM, respectively. TNF cytotoxicity assays were performed in 96-well plates, terminated 18 h post TNF addition and evaluated upon crystal violet staining, solubilization in 33% acetic acid solution and determination of optical density at 570 nm as a test filter and 630 nm as a reference filter.

For MTT assay, briefly, $10^4$ cells/well (5–7 wells/genotype), plated in 96-well plates, were incubated for 24 h and then they were treated with MTT (Sigma-Aldrich; #M5655; final concentration 0.5 mg/ml/well). Plates were incubated for 5 h in the $CO_2$ incubator and the MTT-induced formed formazan is then solubilized in DMSO, and the concentration was determined by optical density at 540 nm.

**Flow cytometric analysis.** Flow cytometric analysis was performed in cells stained with the following Abs: CD90.2 [BioLegend; #105316; 1:300], VCAM-1 [BioLegend; #105712; 1:400], CD45 [BioLegend; #103116; 1:400], ICAM-1 [BD Biosciences; #553253; 1: 400], and CD11b [BD Biosciences; #557397; 1:400] using the FlowJo analysis software.

**Patients and isolation of human SFs.** Synovial tissues were obtained from patients diagnosed with OA or RA undergoing joint replacement surgery at the Schulthess Clinic Zurich. RA patients fulfilled the American Rheumatism Association 1987 revised criteria for the classification of RA. Patient's characteristics are summarized in Supplementary Table 2.

SFs were isolated by digestion with dispase (37 °C, 1 h) and cultured in supplemented DMEM. All reagents were purchased from Life Technologies.

**Cytotoxicity assay in human SFs.** SFs were seeded at a density of $1.5 \times 10^4$ in 96-well plates. After 24 h, cells were treated for 24 h with 10 ng/ml recombinant human TNF-alpha [R&D systems; #210-TA], 50 µM InSolution™ MG-132 [Merck; #474791], 20 µM zVAD, 30 µM Nec-1 and/or 30 µM Nec-1s in duplicates. The evaluation of viability was performed as described for murine SFs.

**Immunoblotting.** Samples were collected in RIPA buffer, containing 1% Triton X-100, 0.1% sodium dodecyl sulfate (SDS), 150 mM NaCl, 10 mM Tris HCl, pH 7.4, 1

mM EDTA, protease inhibitors [Roche] and phosphatase inhibitors [Sigma-Aldrich], separated by SDS/polyacrylamide gel electrophoresis (8–12.5%), transferred to nitrocellulose membranes (Millipore) and probed with the following Abs: IKK2 (#8943 or 2684; 1:1000), pJNK1/2 (#4668; 1:1000), cleaved caspase-3 (CC3; #9661; 1:1000), PARP-1 (#9532; 1:1000), SOD2 (#13194; 1:1000), FLIP (#3210; 1:1000), XIAP (#2042; 1:1000); ikB (#sc-371; 1:1000), tubulin (#sc-9104; 1:2000), actin (#sc-1615; 1:2000) [Santa Cruz Biotechnology, Inc.]; XIAP (#ADI-AAM-050-E; 1:500), cIAP1 (#ALX-803–335-C100; 1:500), Ripk3 (#ADI-905-242-100; 1:1000) [ENZO]; Mlkl [Millipore; #MABC604; 1:1000]; and pMlkl [Abcam; #ab196436; 1:2000]. Relative quantitation of blots was performed either with the ImageLab (BioRad) or Gene Sys (SYNGENE) software.

For human samples, probing were performed with the following Abs: tubulin (Abcam; #ab21057; 1:2000), MLKL (Millipore; #MABC604; 1:1000), and RIP3 (Santa Cruz Biotechnology, Inc.; sc-374639; 1:1000). Relative quantitation has been performed with the ImageJ software.

**Statistical analysis**. Data are presented as mean ± SE. Student's *t*-test (parametric, unpaired, two-sided) was used for evaluation of statistical significance (SigmaPlot 11 and GraphPad 6 software). *P*-values < 0.05 were considered significant.

**Study approval**. Experiments with mice were performed in the conventional unit of the animal facilities in Biomedical Sciences Research Center (BSRC) "Alexander Fleming" under specific pathogen–free conditions, in accordance with the guidance of the Institutional Animal Care and Use Committee of BSRC "Alexander Fleming" and in conjunction with the Veterinary Service Management of the Hellenic Republic Prefecture of Attika/Greece. Experiments were monitored and reviewed throughout its duration by the respective Animal Welfare Body for compliance with the permission granted. Experiments with human SFs were approved by the Swiss Ethical Commission and patients signed an informed consent form.

**Data availability**. The data that support the findings of this study are available from the corresponding authors upon reasonable request.

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

## Acknowledgements

The authors thank Panos Athanasakis, Spyros Lalos, Dimitra Papadopoulou and Anna Katevani for excellent technical assistance. Authors also thank Christoph Kolling (Schulthess Clinic Zurich, Zurich, Switzerland) for collecting the synovial tissue samples and patient's information. We also thank Vishva Dixit (Genetech, USA) for providing *Ripk3*-deficient animals and Claude Libert (VIB, Ghent, Belgium) for the recombinant murine TNF. This work has been supported by the grant from the Stavros Niarchos Foundation to the Biomedical Sciences Research Center "Alexander Fleming" (startup fund to M.A.), as part of the Foundation's initiative to support the Greek research center ecosystem, and IMI-funded project BeTheCure (BTCure, 115142–2), Greek GSRT project INNATE FIBLROBLAST (ERC06, co-financed by the ESF and NSRF 2007–2013) and Advanced ERC grant MCs-inTEST (340217) to G.K. The authors also wish to thank the InfrafrontierGR infrastructure (co-financed by the ERDF and NSRF 2007-2013) for excellent mouse hosting, transgenesis, flow cytometry and microCT facilities.

## Author contributions

All authors read the manuscript and provided feedback. M.A. designed the study, conducted the experiments, collected and analysed the data and prepared the manuscript. C.O. conducted experiments and collected data. M.P. edited the manuscript. G.K. designed the study, analysed the data and edited the manuscript.

## Additional information

**Competing interests:** The authors declare no competing financial interests.

