## [Peer Review File · Nature Communications]

Reviewers' comments:

Reviewer #1 (Remarks to the Author):

The authors address the interaction between signaling pathways downstream of TNFR activation, namely NFkB-dependent inflammation and RIPK3-dependent necroptosis in synovial fibroblasts in the context of rheumatoid arthritis. The experiments are well-conducted and clearly presented. My comments are as follows:

Major comments:

1. Defining the arthritogenic role of SF is surprising and appreciated, but was already reported in the previous J Exp Med paper (9 years ago), so lacks some novelty. Addressing the same aspect with another set of KO mice is a logic follow up project but essentially confirmatory. In addition, the study is much focused in the detail level of signaling events of a pathway that is successfully blocked since 15 years in daily practice. While it is appreciated to place basic science into a translational context it remains unclear to what this project should lead as current RA treatments block the pathway for upstream at the ligand level. It would be interesting to look more in the TNFR-independent forms of NFkB and RIP3 activation to see if dual blockade can elicit therapeutic effects beyond TNF blockade.
2. To conclude on apoptosis requires staining for caspase 3.
3. To conclude on necroptosis requires the use of MLKL KO mice or the use of specific inhibitors of the necroptosis pathway. RIPK3 is not specific for necroptosis but also involved in other signaling pathways.
4. Nec-1 is problematic as inhibitor. Please provide data also with Nec-1s.

Minor comments

1. The introduction is way too long and exhausts the reader before even arriving at the results section. No need to provide a review of all known signaling interactions. Shorten by at least 50%. Also delete the last paragraph on what you found and what it means. This we read already in the abstract and will read in the summary. The introduction serves only one purpose, i.e. to provide the rationale for the hypothesis, the most important element of every research project. The hypothesis is a statement that can be verified or falsified placed as the last sentence of the introduction.

Reviewer #2 (Remarks to the Author):

This paper is an interesting extension of previous observations that TNF receptor signaling specifically in synovial fibroblasts through the p55 TNF-R is critical in various arthritis models. The data reinforce the concept that events downstream of TNF-R are notoriously complex, stimulating either cell activation versus cell death (and more than one type of cell death), and with differences in TNF effects in acute versus chronic exposure of FLS to TNF. I have a few suggestions.

- 1- The authors freely admit that this paper is a partial dissection of this complexity rather than a test of specific hypotheses. Still, the flow of the data would be easier for the reader to follow if the paper was reframed as a logical sequence of tests of specific hypotheses.
- 2- The introduction is too long. It could be cut by 20% while focusing only on arthritis-related aspects of TNF effects.
- 3- The genes for which expression is examined in Figure 4C could be expanded to include the genes whose protein products expressed by FLS are most relevant to joint destruction: MMP14 (the FLS MMP most important in cartilage invasion), and RANK ligand, involved in osteoclast activation.
- 4- Relevance of the findings to human RA would be enhanced by extension of the work to human synovial fibroblasts. This would require knockdown strategies and is mostly beyond the scope of this paper.
- 5- Implications of this work for treatment of RA seem overstated, as pharmacologic agents that

affect TNF-R signaling would have no specificity for synovial fibroblasts, whereas the data in this paper are specific for that cell type.

Reviewer #3 (Remarks to the Author):

This manuscript demonstrates the reduction of hTNF arthritis and CAIA in mice with the mesenchymal deletion of IKK2 (MD-IKK2). When the hTNF-Tg-IKK2 deleted mice were crossed with RIPK3 mice the arthritis was prevented. TUNNEL positivity is seen in MD-IKK2 hTNF mice, and this is ablated in mice with deficient p55TNFR1, supporting the role of TNF induced apoptosis in vivo. Ex vivo studies are presented in an effort to define the pathways that are affected in the mice. The studies extend those by this group that showed the role of mesenchymal TNFR1 in the pathogenesis of arthritis in TNFARE mice. The strength of the manuscript is identifying the role of mesenchymal IKK2 and RIPK2 in the pathogenesis of hTNF-induced arthritis.

Concerns:

- 1) I found the manuscript, as presented difficult to follow. It might be helpful to organize the figures and supplemental figures differently to make the presentation flow more smoothly. I believe the introduction could be shortened and focused.
- 2) The use of the term SF-KO to describe the in vivo deletion of IKK should be documented. While the promoter used may indeed effect SFs other cell types are likely affected. The use of mesenchymal as in their earlier JEM article seems more appropriate, pending further documentation.
- 3) The data concerning the role of NF-kB activation in SF survival in the presence of TNF are not novel, and similar data were published over 15 years ago.
- 4) The ex vivo studies employing hTNF SFs was complicated to interpret, especially Figure 4. How much TNF is in the cultures? Is the effect due to changes after the initiation of culture or in vivo? What happens, under basal conditions, if you inhibit TNF?
- 5) The data in Figure 4D have been shown in other cell types.
- 6) The MTT assay is not a measure of mitochondrial activity, but a measure of proliferation or survival.
- 7) The data in a number of the figures is not convincing and should be documented by quantitation. Specifically: Figure 3C, Figure 4A, Figure 4E, supplemental figure 3C.
- 8) Quantitation of the TUNNEL results should also be presented, not just representative figures.

Minor concerns:

- 1) The mouse used for supplemental Figure 3B should be explained briefly.
- 2) In the histology, clarify proliferation vs synovitis (Figure 2 vs supplemental Figure 5). Do the authors mean inflammation?

Reviewer #1 (Remarks to the Author):

Major comments:

1. Defining the arthritogenic role of SF is surprising and appreciated, but was already reported in the previous J Exp Med paper (9 years ago), so lacks some novelty. Addressing the same aspect with another set of KO mice is a logic follow up project but essentially confirmatory. In addition, the study is much focused in the detail level of signaling events of a pathway that is successfully blocked since 15 years in daily practice. While it is appreciated to place basic science into a translational context it remains unclear to what this project should lead as current RA treatments block the pathway for upstream at the ligand level. It would be interesting to look more in the TNFR-independent forms of NFkB and RIP3 activation to see if dual blockade can elicit therapeutic effects beyond TNF blockade.

We comprehend the remark of the reviewer. We apologize we didn't properly describe the novelty of our current results. In an effort to clarify our new findings, herein we are not only expanding the previous knowledge on the sufficiency of mesenchymal TNFRI. We considered the TNFRI requirements as an important pre-requisite to elucidate the downstream signaling events in acute and chronic TNF conditions. In this regard, we had to unravel whether a common cellular basis exists in TNF-mediated inflammatory and destructive arthritides, independently of stimuli instigating pathology (stromal (as in hTNFtg), innate (as in TNF^{ARE}) or adaptive (as in CAIA)). So this study establishes the role of fibroblasts in supporting and regulating inflammatory responses and provided the necessary cellular dissection for the following signaling studies in an *in vivo* context.

An *in vivo* cell-specific genetic targeting of *ikk2* locus in course of arthritis has never been reported in the literature regardless of the individual studies employing inhibitors or adenoviral-mediated intra-articular targeting. By our approach, we originally noticed the differential mesenchymal inflammatory and death responses caused by chronic or acute pathogenic TNF signals which triggered us to analyze the causative underlying molecular mechanisms. In this regard, we originally provide negative evidence on the role of *Ripk3* inhibition in hTNFtg modeled disease, in sharp contrast to the improvement seen by the concomitant mesenchymal targeting of *ikk2*. Thus, our findings provide a new concept for potential therapeutic approaches in acute (early inflammatory) versus late (chronic inflammatory) arthritic patients beyond the efficacy of the well-established anti-TNF Ab-based treatments.

Regarding the second concern, it would be indeed interesting to examine other, non-TNF operating pathways in arthritis and in a mesenchymal-mediated NFkB/RIPK3 dependent mode. However, most of highly employed models in arthritis depend on TNF[1]. So identifying mechanisms beyond TNF involvement in arthritis is a real challenge. Such approach would require dissection of the cellular players in the remaining non-TNF-dependent models of disease, and deep knowledge of the underlying cellular pathogenetic mechanisms. Interestingly, a recent elegant publication has demonstrated the role of RIPK3 in KRN-serum for transferring arthritic disease, a model minimally dependent on TNF, and

fully dependent on innate compartment and IL1b secretion. The authors correlated the observed accelerated resolution phase of disease under RIPK3-deficiency (regardless of the similarities in initiation phase with the control WT-treated mice) with a mechanism which involves a TLR4/TRIF/RIPK3-mediated NLRP3 inflammasome activation and IL-1b secretion by macrophages, but not necroptosis [2]. We did discuss this study in Discussion part. In any case, this and our findings point to the differences seen among models, depending on their cellular and molecular requirements indicating that it is interesting, yet impossible to test such important hypotheses and perform comparisons in different *in vivo* systems within the current study.

2. To conclude on apoptosis requires staining for caspase 3.

We fully appreciate the reviewer's comment that additional *in vivo* evidence for the detection of apoptosis is required. For this reason we performed immunohistochemistry for the detection cleaved caspase-3 in joints of all related genotypes and arthritic conditions. The representative results are incorporated in Fig 3A. Notably, we detected qualitative, apart from quantitative, differences in the localization of CC3+ cells within the synovium: the CC3 detection was mostly apparent in the lining SFs of both naïve and CAIA-treated mesenchymal *ikk2KO* mice in contrast to hTNFtg mesenchymal *ikk2KO* mice. In chronic arthritis, the mesenchymal *ikk2* deficiency leads to a sub-lining SF apoptosis in a few cells indicating that different responses within synovium might take place in response to chronic versus acute TNF conditions *in vivo*.

3. To conclude on necroptosis requires the use of MLKL KO mice or the use of specific inhibitors of the necroptosis pathway. RIPK3 is not specific for necroptosis but also involved in other signaling pathways.

We fully agree with the reviewer. Emerging evidence regarding *ripk3* indeed indicates the cell-specific context of *ripk3* involvement in death and other signaling pathways. For an *in vivo* approach, the time restriction in immediate access to *Mlkl-KO* mice and the generation of a quadruple genotype (even with direct CRISRR-Cas targeting) prevent us from providing the involvement of *Mlkl* in the current study. Thus, we changed the related statements in the text. Moreover, given our current results, we plan to address the relative contribution and role of necroptosis and apoptosis in the pathology in future studies using *MLKL* and other related targeted mice.

4. Nec-1 is problematic as inhibitor. Please provide data also with Nec-1s.

We performed some additional *ex vivo* experiments with more specific inhibitors. *Nec1s* (RIPK1 inhibitor[3]) and *GW806742X* (*MLKL* inhibitor[4]) are currently the most specific, commercially available small molecule inhibitors for testing aspects of death pathways *ex vivo*. The results were incorporated in Fig 4D (replacing the previous version of the experiment with *zVAD/Nec1* treatments) and they confirm the *Ripk1* and *Mlkl* requirements in synovial demise *ex vivo*. Even though some recent evidence suggests that the *Ripk1* kinase activity could be also implicated in processes beyond necroptosis [5, 6] and the *GW806742X*

inhibitor could also affect the (human) RIPK1/3 activity[7], our overall *ex vivo* and *in vivo* results support the involvement of RIPK1/3-mediated death entities in SFs.

Minor comments

1. The introduction is way too long and exhausts the reader before even arriving at the results section. No need to provide a review of all known signaling interactions. Shorten by at least 50%. Also delete the last paragraph on what you found and what it means. This we read already in the abstract and will read in the summary. The introduction serves only one purpose, i.e. to provide the rationale for the hypothesis, the most important element of every research project. The hypothesis is a statement that can be verified or falsified placed as the last sentence of the introduction.

We thank the reviewer for his useful suggestions regarding the improvement of our text. We agree that the introduction was long and we re-arrange the text according to the recommendations.

References

1. Apostolaki, M., et al., *Cellular mechanisms of TNF function in models of inflammation and autoimmunity*. Curr Dir Autoimmun, 2010. **11**: p. 1-26.
2. Lawlor, K.E., et al., *RIPK3 promotes cell death and NLRP3 inflammasome activation in the absence of MLKL*. Nat Commun, 2015. **6**: p. 6282.
3. Degtarev, A., et al., *Chemical inhibitor of nonapoptotic cell death with therapeutic potential for ischemic brain injury*. Nat Chem Biol, 2005. **1**(2): p. 112-9.
4. Conrad, M., et al., *Regulated necrosis: disease relevance and therapeutic opportunities*. Nat Rev Drug Discov, 2016. **15**(5): p. 348-66.
5. Daniels, B.P., et al., *RIPK3 Restricts Viral Pathogenesis via Cell Death-Independent Neuroinflammation*. Cell, 2017. **169**(2): p. 301-313 e11.
6. Newton, K., et al., *RIPK3 deficiency or catalytically inactive RIPK1 provides greater benefit than MLKL deficiency in mouse models of inflammation and tissue injury*. Cell Death Differ, 2016. **23**(9): p. 1565-76.
7. Yan, B., et al., *Discovery of a new class of highly potent necroptosis inhibitors targeting the mixed lineage kinase domain-like protein*. Chemical Communications, 2017. **53**(26): p. 3637-3640.

Reviewer #2 (Remarks to the Author):

This paper is an interesting extension of previous observations that TNF receptor signaling specifically in synovial fibroblasts through the p55 TNF-R is critical in various arthritis models. The data reinforce the concept that events downstream of TNF-R are notoriously complex, stimulating either cell activation versus cell death (and more than one type of cell death), and with differences in TNF effects in acute versus chronic exposure of FLS to TNF. I have a few suggestions.

1- The authors freely admit that this paper is a partial dissection of this complexity rather than a test of specific hypotheses. Still, the flow of the data would be easier for the reader to follow if the paper was reframed as a logical sequence of tests of specific hypotheses.

We agree with the reviewer and we comprehend and share his concerns. We adjusted the text accordingly.

2- The introduction is too long. It could be cut by 20% while focusing only on arthritis-related aspects of TNF effects.

We appreciate the effort and the comments of the reviewer to improve the presentation of our study and we followed his recommendation in rearranging the introduction text.

3- The genes for which expression is examined in Figure 4C could be expanded to include the genes whose protein products expressed by FLS are most relevant to joint destruction: MMP14 (the FLS MMP most important in cartilage invasion), and RANK ligand, involved in osteoclast activation.

We performed the requested experiment and we incorporated the results in Figure 4D. Briefly, MMP14 and RankL expression is regulated by *ikk2*-mediated signals in acute TNF triggering of non-arthritic *ikk2ko* SFs. However, only MMP14 is significantly downregulated in arthritic hTNF-Tg *ikk2ko* SFs, supporting the absence of erosions in the tissue of the animals. Interestingly, TNF-triggering does not cause further deregulation of MMP14 in hTNF-Tg *ikk2KO* SFs. RankL does not seem severely affected by the absence of *ikk2* in arthritic hTNF-Tg *ikk2ko* SFs whereas further inflammatory stimulation fails to upregulate the expression in the same cells.

4- Relevance of the findings to human RA would be enhanced by extension of the work to human synovial fibroblasts. This would require knockdown strategies and is mostly beyond the scope of this paper.

We understand the concerns of the reviewer and the editors' requirements, and we tried to address them by performing experiments in human cells affecting pathways relevant to the murine SFs (Supplementary Figure 7).

As previously reported, *ikk2* inhibition does not potently change anti-apoptotic gene expression so as to provoke death in RA- or OA-SFs upon TNF treatment and we indeed confirm this finding (data not shown). Thus we couldn't implicate the death pathway activation under this experimental setting. To identify how disturbances in survival pathways in human cells share similarities with our murine system, we took advantage of other inhibitors, such as the MG132 (carbobenzoxy-Leu-Leu-leucinal), a peptide aldehyde which effectively blocks the proteolytic activity of the 26S proteasome complex thus affecting multiple pathways including NFkB, and inducing apoptotic cell death[1, 2]. In collaboration with Caroline Ospelt, we applied the MG132 inhibitor on RA-SFs and OA-SFs along with the combination of TNF and inhibitors for caspase and RIPK1 inhibition. We noticed that upon

TNF stimulation in the MG132-treated OASFs or RASF samples, the caspase or RIPK1 inhibition marginally reverses death, abrogating any favoring for any of the two death pathways in any cell type. Contrary to single treatments, a complete survival benefit was observed by the combination of the RIPK1 and pan-caspase inhibitor (Suppl. Fig. 7C). Interestingly, by looking at individual sample responses (Suppl. Fig. 7D), we noticed that 2 out of 5 samples (40%) from both OA and RA patients were partially rescued by each inhibitor treatment suggesting that the TNF/MG132-mediated effects are not limited to a caspase mediated death but also includes a RIPK1-mediated demise in the SFs of some patients. We also confirmed the RIPK1 specificity by applying Nec1s inhibitor in responsive samples which indeed showed a similar pattern of responses (Suppl. Fig. 7E).

Even though the human and murine experimental settings differs thus preventing us from directly correlating *ikk2* inhibition with cell death induction mechanisms in human arthritic conditions, the observed death responses in human cells are in accordance with our observation in murine *ikk2*-ko SFs, implicating both caspase-dependent and -independent demise being active in SFs of some patients with OA or RA upon survival signaling inhibition. The presence of inflammatory environment in OA [3-5] may account for the absence of differences in death responses between OA- and RA-SFs, also indicating that SFs from healthy individuals would be the ultimate control. Moreover, NFkB inhibition has been also proposed for the treatment of OA [5, 6]. Other possible explanation could be that MG132 inhibitor, beside NFkB, is targeting the ubiquitin-related degradation processes which might affect Ripk1 and/or Ripk3 functions in the formation of a fully-functional necrosome [7, 8]. More detailed analysis is required towards these directions in order to provide sufficient mechanistic insights and unfortunately such analysis is beyond the scope of this study. Nonetheless, we believe that our results indicates the involvement of differential death responses in the absence of intact survival signaling in human SFs. Interestingly, since proteasomal inhibition targeting NFkB and other survival pathways has been also proposed for the treatment of RA and OA[9], our human-related evidence suggests the consideration of Ripk1-mediated responses (including necroptotic death) which may affect cell behavior and disease outcome similar to our observations in the murine settings.

5- Implications of this work for treatment of RA seem overstated, as pharmacologic agents that affect TNF-R signaling would have no specificity for synovial fibroblasts, whereas the data in this paper are specific for that cell type.

Our suggestions could sound challenging based on the current availability of therapeutic tools. However, we want to clarify that herein we provide *in vivo* mechanistic insights on the physiology and pathogenicity of the signaling pathways in SFs upon acute or chronic inflammatory TNF challenge. Our evidence advocates for a cell-specific targeting of the *ikk2*-mediated pathway in combination with RIPK3 systemic inhibition as a proof-of-concept bi-directional therapeutic approach. Unfortunately, little evidence exists on the development of cell-specific targeting approaches that could have enhanced our experimental evidence if they would have been available (as eg. the Endothelial-directed siRNA mediated targeting [10]). However, previous studies on intra-articular, yet non cell-specific, adenoviral-based targeting approaches in mice led to promising results *in vivo*[11, 12] indicating that our

suggested methodology (synovial ikk2 inhibition combined with systemic ripk3 blockade) could be achieved upon the development of synovial-specific targeting technologies in the future. Furthermore, RIPK3 is activated through RIPK1 and the pharmacological targeting of the upstream RIPK1-kinase activity is currently passed to the Phase II in clinical trials for RA and other diseases. If future studies will address a similar-to-RIPK3 role for RIPK1 kinase in SFs and TNF-mediated arthritis, a potential therapeutic benefit could be tested by a combination of SF-specific NF κ B inhibition along with the RIPK1 inhibitor.

References

1. Kato, M., et al., *Dual role of autophagy in stress-induced cell death in rheumatoid arthritis synovial fibroblasts*. Arthritis Rheumatol, 2014. **66**(1): p. 40-8.
2. Lee, D.H. and A.L. Goldberg, *Proteasome inhibitors: valuable new tools for cell biologists*. Trends Cell Biol, 1998. **8**(10): p. 397-403.
3. Sokolove, J. and C.M. Lepus, *Role of inflammation in the pathogenesis of osteoarthritis: latest findings and interpretations*. Therapeutic Advances in Musculoskeletal Disease, 2013. **5**(2): p. 77-94.
4. Rahmati, M., A. Mobasheri, and M. Mozafari, *Inflammatory mediators in osteoarthritis: A critical review of the state-of-the-art, current prospects, and future challenges*. Bone, 2016. **85**: p. 81-90.
5. Mobasheri, A., *The Future of Osteoarthritis Therapeutics: Emerging Biological Therapy*. Current Rheumatology Reports, 2013. **15**(12): p. 385.
6. Kenneth, B.M., et al., *NF- κ B Signaling: Multiple Angles to Target OA*. Current Drug Targets, 2010. **11**(5): p. 599-613.
7. de Almagro, M.C., et al., *Cellular IAP proteins and LUBAC differentially regulate necrosome-associated RIP1 ubiquitination*. Cell Death Dis, 2015. **6**: p. e1800.
8. Moriwaki, K. and F.K.-M. Chan, *Regulation of RIPK3- and RHIM-dependent Necroptosis by the Proteasome*. The Journal of Biological Chemistry, 2016. **291**(11): p. 5948-5959.
9. Roman-Blas, J.A. and S.A. Jimenez, *NF- κ B as a potential therapeutic target in osteoarthritis and rheumatoid arthritis*. Osteoarthritis Cartilage, 2006. **14**(9): p. 839-48.
10. Dahlman, J.E., et al., *In vivo endothelial siRNA delivery using polymeric nanoparticles with low molecular weight*. Nature nanotechnology, 2014. **9**(8): p. 648-655.
11. Tas, S.W., et al., *Amelioration of arthritis by intraarticular dominant negative Ikk beta gene therapy using adeno-associated virus type 5*. Hum Gene Ther, 2006. **17**(8): p. 821-32.
12. Tas, S.W., et al., *Local treatment with the selective IkkappaB kinase beta inhibitor NEMO-binding domain peptide ameliorates synovial inflammation*. Arthritis Res Ther, 2006. **8**(4): p. R86.

Reviewer #3 (Remarks to the Author):

This manuscript demonstrated the reduction of hTNF arthritis and CAIA in mice with the mesenchymal deletion of IKK2 (MD-IKK2). When the hTNF-Tg-IKK2 deleted mice were crossed with RIPK3 mice the arthritis was prevented. TUNNEL positivity is seen in MD-IKK2 hTNF mice, and this is ablated in mice with deficient p55TNFR1, supporting the role of TNF induced apoptosis *in vivo*. *Ex vivo* studies are presented in an effort to define the pathways that are affected in the mice. The studies extend those by this group that showed the role of mesenchymal TNFR1 in the pathogenesis of arthritis in TNFARE mice. The strength of the manuscript is identifying the role of mesenchymal IKK2 and RIPK3 in the pathogenesis of hTNF-induced arthritis.

Concerns:

1) I found the manuscript, as presented difficult to follow. It might be helpful to organize the figures and supplemental figures differently to make the presentation flow more smoothly. I believe the introduction could be shortened and focused.

We re-organized parts of the manuscript including introduction and we tried to adjust the figures to the new format as much as possible. We hope that these changes will please the reviewer.

2) The use of the term SF-KO to describe the *in vivo* deletion of IKK should be documented. While the promoter used may indeed affect SFs other cell types are likely affected. The use of mesenchymal as in their earlier JEM article seems more appropriate, pending further documentation.

We thank the reviewer for the recommendation, thus we changed the SF-KO to MS-KO abbreviation.

3) The data concerning the role of NF-kB activation in SF survival in the presence of TNF are not novel, and similar data were published over 15 years ago.

As we discussed a similar argument by Reviewer 1 (1st comment), NFkB was indeed analyzed in previous studies. However, there was no evidence on a cell-specific mode of action in a murine chronic or acute arthritic model, and several studies indicated in the Introduction part presented differential responses of *ikk2*-targeted cells depending on the stimuli and the context. Thus, not only we confirm existing evidence, but we provide mechanistic insights on the role of *ikk2*-mediated signals particularly in SFs *in vivo*, on both acute/self-resolving and chronic arthritic conditions. We additionally provide the *ikk2*-mediated effects in the course of disease and we dissect the molecular requirements (RIPK3-mediated) relevant to the physiology and pathophysiology of SF behavior in an *ex vivo* and *in vivo* context.

Additionally, this is the first time to show that RIPK3 blockade did not affect TNF-mediated arthritis; however a combination of cell-specific NFkB inhibition with systemic use of a RIPK3-mediated signaling inhibition could act beneficially in the arthritic context.

4) The *ex vivo* studies employing hTNF SFs was complicated to interpret, especially Figure 4.

We apologize we didn't include much detail on our experimental settings. Therefore we added all related information in the text of results and the respective Figure legends.

A. How much TNF is in the cultures?

hTNF-Tg SFs secrete TNF *ex vivo* as this has been evaluated by ELISA (almost 1ng/ml/ 10⁵ cell) (Fig 4B, upright panel).

B. Is the effect due to changes after the initiation of culture or *in vivo*?

As exhibited in Fig 3, Suppl. Fig 3 and in related text, the *ikk2* deletion *in vivo* (*ColVICre ikk2^{ff}* mice) abrogated the expansion of primary SF cultures *ex vivo*, in a p55TNFR-dependent mode (note the successful deletion of *ikk2* under TNFR1-deficient conditions, Suppl. Fig3B). However, targeting *ikk2* in SFs *ex vivo* upon the expansion of *ikk2^{ff}* cultures (by the application of a cell-permeable Cre peptide), the deletion of *ikk2* as well as the expansion of newly-generated *ikk2*-KO cells was successful. Considering the detection of death events *in vivo*, we can suggest that insufficient survival of *ikk2* KO SFs generated *in vivo* could not provide sufficient cell numbers to expand cultures for the appropriate analysis *ex vivo*.

C. What happens, under basal conditions, if you inhibit TNF?

We provide evidence regarding inhibition of TNF signaling by the genetic p55TNFR1-deficiency effect on *ColVICre ikk2^{ff}*-derived SF cultures. As previously mentioned, we detect the successful deletion of *ikk2* and expansion of SFs *ex vivo* under TNFR1-deficient conditions (*ColVICre ikk2^{ff} p55TNFR^{-/-}*, Suppl. Fig3B). Moreover, we evidence that the *ikk2*/p55TNFR deficient SFs are defective in signaling through *ikk2* by the examination of these cells upon challenging with LPS or IL1 (Suppl. Fig3D).

5) The data in Figure 4D have been shown in other cell types.

We agree with the reviewer that some published evidence regarding similar responses exist. However the murine evidence concerns mainly either *ikk2*KO MEFs (non-differentiated fibroblasts) or L292 mouse fibroblast cell line and their responses are closer to hTNFTg *ikk2*ko rather than *ikk2*ko SFs in culture. So, herein, we report on the differences noted between *ikk2*ko versus hTNFTg *ikk2*ko SF, a specialized and disease-promoting cell type compared to MEFs or L292 cells, and we translate the *ex vivo* differential responses into a disease context.

6) The MTT assay is not a measure of mitochondrial activity, but a measure of proliferation or survival.

We thank the reviewer for the comment and the updated information regarding MTT; we changed the relative statement.

7) The data in a number of the figures is not convincing and should be documented by quantitation. Specifically: Figure 3C, Figure 4A, Figure 4E, supplemental figure 3C.

We agree with the reviewer, so we provide the relative evidence on the quantitation of the blots in graphs (within Fig 3D (previous marking Fig 3C); Suppl. Fig 4 for Fig 4C and F (previous marking Fig 4A and E blots); within Suppl. Fig. 3D (previous marking Suppl. Fig 3C))

8) Quantitation of the TUNEL results should also be presented, not just representative figures.

We followed the recommendation of the reviewer and we added the quantitation of TUNEL positive cells in the synovium area surrounding tallus (tallus/tibia joint and tallus/calcaneus) in Fig 3B.

Minor concerns:

1) The mouse used for supplemental Figure 3B should be explained briefly.

We added some more information and explanations regarding the experiment presented in the figure 3C (previously marked as 3B). Briefly, we used mice with a reporter gene, inserted in ROSA locus [1]. In non-Cre expressing conditions, Tomato gene reporter is expressed in all cells. Upon Cre mediated recombination, the floxed stop codon is removed so the GFP expression is initiated, marking green all the Cre-expressing cells. In the presented experiment, SFs derived from *ColVICre ikk2^{ff} Rosa^{mT/mG}* and control *ColVICre Rosa^{mT/mG}* were isolated and expanded in order to instantly detect the percentage of Cre-expressing cells by FACS. We noticed that in the presence of functional *ikk2* (*ColVICre Rosa^{mT/mG}* SFs), the percentage of Cre-expressing SFs (CD90^{high}) in the culture is almost 70%, exhibiting the recombination efficiency of the ColVI-Cre mouse in SFs *ex vivo*. Under the same culturing conditions, the concomitant *ikk2* deficiency (*ColVICre ikk2^{ff} Rosa^{mT/mG}* SFs) reduced the percentage of Cre-expressing cells in the culture to 25%, implicating survival or/and expanding deficits in *ikk2*-targeted GFP-expressing cells.

2) In the histology, clarify proliferation vs synovitis (Figure 2 vs supplemental Figure 5). Do the authors mean inflammation?

We thank the reviewer for noticing the entry of a new term in our histopathological presentation. However this was stated by mistake since synovial hyperplasia and synovitis is evaluated as one entity in all histopathological evaluations herein, as this was also described in our scoring system indicated in the Materials and Methods. So the Synovitis is changed to Synovial hyperplasia/inflammation to uniformly state our evaluations.

References

1. Muzumdar, M.D., et al., *A global double-fluorescent Cre reporter mouse*. Genesis, 2007. **45**(9): p. 593-605.

REVIEWERS' COMMENTS:

Reviewer #2 (Remarks to the Author):

The issues that I raised have been addressed - thank you.

Reviewer #3 (Remarks to the Author):

The authors have thoroughly and thoughtfully responded to the reviewers comments. However, the work addressing in NF-kB inhibition in vitro employing synovial fibroblasts remains at best incremental. Nonetheless there are substantial novel insights that have been highlighted in the revised manuscript.

I suggest:

- 1) more clearly define the effects of IKK2 deletion on synovial fibroblasts.
- 2) revise the second to the last sentence in the abstract to make it clearer.

Reviewer #4 (Remarks to the Author):

The Editors did not request a full review, but instead asked for some specific comments regarding whether the authors have satisfactorily addressed reviewer 1 requests.

In the first major comment, referee #1 asked:

"It would be interesting to look more in the TNFR-independent forms of NFkB and RIP3 activation to see if dual blockade can elicit therapeutic effects beyond TNF blockade."

The authors responded:

"An in vivo cell-specific genetic targeting of *ikk2* locus in course of arthritis has never been reported in the literature regardless of the individual studies employing inhibitors or adenoviral-mediated intra-articular targeting."

To the best of my knowledge, this is correct. More importantly, the authors argue:

"In this regard, we originally provide negative evidence on the role of Ripk3 inhibition in hTNFtg modeled disease, in sharp contrast to the improvement seen by the concomitant mesenchymal targeting of *ikk2*. Thus, our findings provide a new concept for potential therapeutic approaches in acute (early inflammatory) versus late (chronic inflammatory) arthritic patients beyond the efficacy of the well-established anti-TNF Abbased treatments."

Indeed, the role of RIPK3 in this particular setting has never been addressed before. With respect to the Lawlor et al. paper, the response of the authors raises an important point that has not been investigated wither by Lawlor et al or by themselves. This point is of critical importance, because – as the authors correctly mention – RIPK3 as a kinase can be responsible for multiple outcomes on the immune system. The conclusion of "necroptosis" to be the driver of this disease, however, would require MLKL-deficient mice that are not included in this study. Therefore, I suggest that the authors refer to "RIPK3-dependent inflammation" rather than "necroptosis" as the driver of this disease.

2. "To conclude on apoptosis requires staining for caspase 3."
This issue has been completely addressed.

3. "To conclude on necroptosis requires the use of MLKL KO mice or the use of specific inhibitors of

the necroptosis pathway. RIPK3 is not specific for necroptosis but also involved in other signaling pathways."

This is a very valid comment, and the authors adequately respond. The issue here is complex because it would indeed be nice to conclude on necroptosis, but qko should not be generated to withhold this paper from being published for at least 2 years. This would significantly inhibit the field. However, as stated above, a careful interpretation of the RIPK3-dependent inflammation rather than necroptosis should be performed throughout the paper.

4. "Nec-1 is problematic as inhibitor. Please provide data also with Nec-1s."

Referee #1 raised an important point which has been adequately addressed by the use of more specific inhibitors (Fig. 4D).

All minor comments were adequately corrected.

In summary, I conclude that the comments of referee #1 have been intensively taken into consideration, additional experiments have been performed and the criticism has led to a better quality of the manuscript. I therefore recommend this paper for publication.

Reviewer #3 (Remarks to the Author):

The authors have thoroughly and thoughtfully responded to the reviewers comments. However, the work addressing in NF-kB inhibition in vitro employing synovial fibroblasts remains at best incremental. Nonetheless there are substantial novel insights that have been highlighted in the revised manuscript.

I suggest:

- 1) more clearly define the effects of IKK2 deletion on synovial fibroblasts.
- 2) revise the second to the last sentence in the abstract to make it clearer.

We thank once more the reviewer for his/her comments and we corrected the abstract and discussion text to clearly define our experimental evidence and the impact of our results.

Reviewer #4 (Remarks to the Author):

The Editors did not request a full review, but instead asked for some specific comments regarding whether the authors have satisfactorily addressed reviewer 1 requests. In the first major comment, referee #1 asked:

"It would be interesting to look more in the TNFR-independent forms of NFkB and RIP3 activation to see if dual blockade can elicit therapeutic effects beyond TNF blockade."

The authors responded:

"An *in vivo* cell-specific genetic targeting of *ikk2* locus in course of arthritis has never been reported in the literature regardless of the individual studies employing inhibitors or adenoviral-mediated intra-articular targeting."

To the best of my knowledge, this is correct. More importantly, the authors argue: "In this regard, we originally provide negative evidence on the role of Ripk3 inhibition in *hTNFtg* modeled disease, in sharp contrast to the improvement seen by the concomitant mesenchymal targeting of *ikk2*. Thus, our findings provide a new concept for potential therapeutic approaches in acute (early inflammatory) versus late (chronic inflammatory) arthritic patients beyond the efficacy of the well-established anti-TNF Ab based treatments." Indeed, the role of RIPK3 in this particular setting has never been addressed before. With respect to the Lawlor et al. paper, the response of the authors raises an important point that has not been investigated wither by Lawlor et al or by themselves. This point is of critical importance, because – as the authors correctly mention – RIPK3 as a kinase can be responsible for multiple outcomes on the immune system. The conclusion of "necroptosis" to be the driver of this disease, however, would require MLKL-deficient mice that are not included in this study. Therefore, I suggest that the authors refer to "RIPK3-dependent inflammation" rather than "necroptosis" as the driver of this disease.

In our original submission we hastened to state necroptotic responses as a driver of synovitis in *hTNFtg Ikk2^{MS-KO}* mice based on the literature available when the experiments had been performed. We had changed the necroptosis-relevant statements for driving synovitis in the second and the current version of the manuscript (both in the Abstract and Results section). We refer a Ripk3-mediated response. Within Results section, we changed the statements for our *in*

vivo results by mentioning that "Mechanistically, the Ripk3 pathway conditionally controls gene expression and/or the cell death decisions which were observed *ex vivo* in the *hTNFtg* *Ikk2*-deficient SFs thus suggesting that the persisting synovitis in *hTNFtg Ikk2^{Ms-KO}* animals could be provoked by a Ripk3-mediated inflammatory process". We only discuss a possible role of necroptosis in our experimental system based on our results and in correlation with the current literature, in the Discussion part. We hope that the reviewer will agree on this approach.

2. "To conclude on apoptosis requires staining for caspase 3." This issue has been completely addressed.

3. "To conclude on necroptosis requires the use of MLKL KO mice or the use of specific inhibitors of the necroptosis pathway. RIPK3 is not specific for necroptosis but also involved in other signaling pathways."

This is a very valid comment, and the authors adequately respond. The issue here is complex because it would indeed be nice to conclude on necroptosis, but qko should not be generated to withhold this paper from being published for at least 2 years. This would significantly inhibit the field. However, as stated above, a careful interpretation of the RIPK3-dependent inflammation rather than necroptosis should be performed throughout the paper.

4. "Nec-1 is problematic as inhibitor. Please provide data also with Nec-1s." Referee #1 raised an important point which has been adequately addressed by the use of more specific inhibitors (Fig. 4D).

All minor comments were adequately corrected.

In summary, I conclude that the comments of referee #1 have been intensively taken into consideration, additional experiments have been performed and the criticism has led to a better quality of the manuscript. I therefore recommend this paper for publication.

We thank the reviewer for his sheer comments and support for our study.